# SMMILE: An Expert-Driven Benchmark for Multimodal Medical In-Context Learning

**Melanie Rieff**[1*]    **Maya Varma**[2*]    **Ossian Rabow**[2,3]    **Subathra Adithan**[4]    **Julie Kim**[2]
**Ken Chang**[2]    **Hannah Lee**[5]    **Nidhi Rohatgi**[2]    **Christian Bluethgen**[2,6,7]
**Mohamed S. Muneer**[2]    **Jean-Benoit Delbrouck**[2,8,†]    **Michael Moor**[1,†]

[1]ETH Zurich    [2]Stanford University    [3]Lund University
[4]Jawaharlal Institute of Postgraduate Medical Education and Research
[5]UCSF    [6]University of Zurich    [7]University Hospital Zurich    [8]HOPPR

## Abstract

Multimodal in-context learning (ICL) remains underexplored despite significant potential for domains such as medicine. Clinicians routinely encounter diverse, specialized tasks requiring adaptation from limited examples, such as drawing insights from a few relevant prior cases or considering a constrained set of differential diagnoses. While multimodal large language models (MLLMs) have shown advances in medical visual question answering (VQA), their ability to learn multimodal tasks from context is largely unknown. We introduce SMMILE, the first expert-driven multimodal ICL benchmark for medical tasks. Eleven medical experts curated problems, each including a multimodal query and multimodal in-context examples as task demonstrations. SMMILE encompasses 111 problems (517 question-image-answer triplets) covering 6 medical specialties and 13 imaging modalities. We further introduce SMMILE++, an augmented variant with 1038 permuted problems. A comprehensive evaluation of 15 MLLMs demonstrates that most models exhibit moderate to poor multimodal ICL ability in medical tasks. In open-ended evaluations, ICL contributes only an 8% average improvement over zero-shot on SMMILE and 9.4% on SMMILE++. We observe a susceptibility for irrelevant in-context examples: even a single noisy or irrelevant example can degrade performance by up to 9.5%. Moreover, we observe that MLLMs are affected by a recency bias, where placing the most relevant example last can lead to substantial performance improvements of up to 71%. Our findings highlight critical limitations and biases in current MLLMs when learning multimodal medical tasks from context. SMMILE is available at https://smmile-benchmark.github.io.

## 1 Introduction

In-context learning (ICL) has been widely studied as the striking ability of large language models (LLMs) to generalize to new tasks at inference time when provided with a few demonstration examples in their input context, without requiring any parameter updates [5]. Given a set of relevant labeled examples in the input prompt, ICL enables models to flexibly adapt to provided contexts, contributing to applications like retrieval-augmented generation [13, 18, 33, 30] and model personalization.

Although ICL has been predominantly studied in the context of LLMs, recent works have explored extensions of ICL to multimodal settings [1, 16, 15, 17]. Multimodal ICL holds particular promise for the domain of medicine due to the close parallels between ICL and clinical workflows; in real-world medical settings, clinicians are routinely asked to address specialized tasks given knowledge of

---

[*]These authors contributed equally to this work. † Co-senior authors.

39th Conference on Neural Information Processing Systems (NeurIPS 2025) Track on Datasets and Benchmarks.

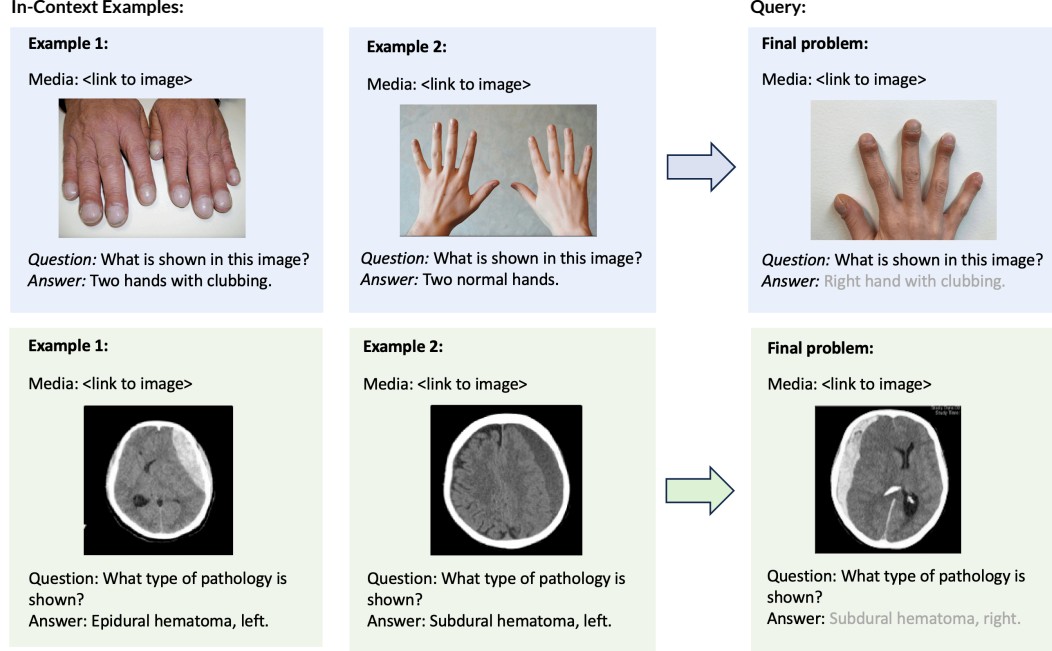

**In-Context Examples:**

**Example 1:**

Media: <link to image>

*Question:* What is shown in this image?
*Answer:* Two hands with clubbing.

**Example 2:**

Media: <link to image>

*Question:* What is shown in this image?
*Answer:* Two normal hands.

**Query:**

**Final problem:**

Media: <link to image>

*Question:* What is shown in this image?
*Answer:* Right hand with clubbing.

**Example 1:**

Media: <link to image>

Question: What type of pathology is shown?
Answer: Epidural hematoma, left.

**Example 2:**

Media: <link to image>

Question: What type of pathology is shown?
Answer: Subdural hematoma, left.

**Final problem:**

Media: <link to image>

Question: What type of pathology is shown?
Answer: Subdural hematoma, right.

Figure 1: Overview of the SMMILE benchmark. In order to test the ability of MLLMs to perform multimodal in-context learning in the medical domain, we curate an expert-annotated dataset consisting of multimodal queries paired with two or more task-specific in-context examples. In contrast to prior few-shot evaluations, our in-context examples are expert-designed demonstrations of the task at hand, rather than randomly retrieved examples.

limited prior examples, such as a few relevant prior cases or a constrained set of differential diagnoses. Models capable of performing multimodal ICL in high-stakes medical settings must be carefully assessed for reliability. Although some prior works have proposed strategies for evaluating the ICL capabilities of multimodal LLMs (MLLMs) in the general domain [34, 4, 6, 37], no benchmarks have been previously developed to systematically evaluate multimodal ICL in the medical domain. Additionally, existing few-shot evaluations in medical settings often randomly select examples rather than focusing on specific task demonstrations [25, 32], which may partially explain why minimal improvements over zero-shot evaluations are often observed.

In this work, we aim to address these challenges by introducing the **S**tanford **M**ultimodal **M**edical **I**n-context **Le**arning (SMMILE) benchmark. Notably, SMMILE is an *expert-driven* benchmark, developed in collaboration with an international team of 11 medical experts. Our contributions are:

- We present SMMILE, the **first expert-driven multimodal ICL benchmark for the medical domain**. Medical experts contributed *problems*, each consisting of (1) a multimodal query to be posed to a MLLM and (2) two or more multimodal in-context examples designed to serve as relevant task demonstrations. In total, SMMILE includes 111 problems encompassing 517 question-image-answer triplets across 6 medical specialties and 13 imaging modalities. We also introduce SMMILE++, an augmented benchmark with 1038 problems designed by permuting the order of in-context examples in SMMILE. Our benchmarks support both open-ended and closed-ended evaluations.

- We evaluate 15 MLLMs on our benchmarks, including both open-source and closed-source models with diverse architectures and model sizes. Model performance is assessed using both automated metrics as well as human expert evaluations. Our results show that existing MLLMs struggle to effectively learn from multimodal in-context examples in the medical setting, with ICL contributing to minimal performance boosts over zero-shot evaluations across most evaluated models. In open-ended settings, even the best-performing models (GPT-4o and Qwen2.5-VL-72B) are only capable of answering approximately half of the questions accurately. These results expose a significant

shortcoming of current MLLMs: although in-context examples are manually designed to serve as effective task demonstrations, MLLMs are unable to accurately learn the task at hand.

- The manually-curated and high-quality nature of the SMMILE benchmark can help reveal insights into how effective in-context examples can be selected for MLLMs. To this end, we perform an in-depth analysis of two critial factors associated with selecting in-context examples. First, we demonstrate that the *quality* of in-context examples is important: the inclusion of just one irrelevant sample in the in-context example list can impair performance. Second, we demonstrate that the *order* of in-context examples matters: all evaluated MLLMs suffer from recency bias, where placing the most relevant in-context examples later in the example list can improve performance.

By highlighting the limited ICL capabilities of current MLLMs, we hope that our benchmark will be a valuable asset for monitoring this critical ability in future MLLMs. Our benchmark can help drive the development of medical MLLMs capable of efficiently learning to perform novel tasks at inference time. SMMILE is available at https://smmile-benchmark.github.io.

**Related Work**   Our work is motivated by prior research on in-context learning, medical MLLMs, and benchmarking efforts. We discuss related works in Appendix Section A.

## 2   SMMILE: Benchmarking Multimodal Medical In-Context Learning

In this section, we describe our expert-guided process for curating data (Section 2.1) as well as provide quantitative analysis of the final SMMILE benchmark (Section 2.2).

### 2.1   Dataset Curation

In order to collect data, we first recruited clinical experts to contribute multimodal ICL problems. In this setting, each problem consists of (1) a *query* to be posed to a MLLM, including a question, an associated non-text media item (e.g. an image), and a ground-truth answer; and (2) two or more *in-context examples*, each including a question, an associated non-text media item, and an answer. Examples are designed to serve as relevant task demonstrations that support a model in learning the task at hand. We provide sample problems from SMMILE in Figure 1.

Experts were given access to a web interface and instructed to create ten problems. Initial recruiting via direct contacting yielded a set of 21 clinical domain experts. Out of this initial set, 11 experts sucessfully complied with the instructions and created problems for the SMMILE dataset. The final set of domain experts includes

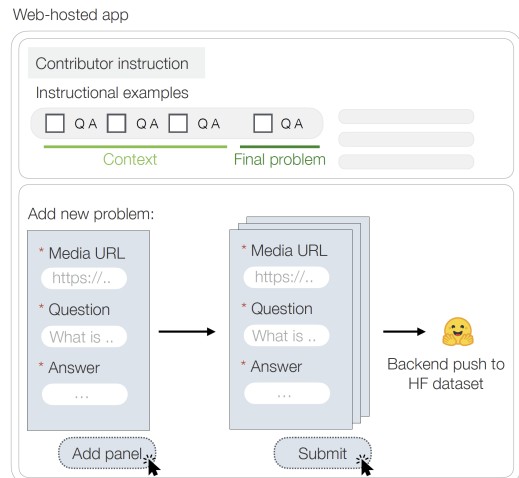

Figure 2: Web interface for data collection.

nine medical doctors and two medical students. The doctors report an average of 6.4 years of clinical experience with specialty expertise in radiology, general medicine, and pathology. For maximal flexibility, we instructed the clinical contributors to create problems by means of writing text and providing URLs to publicly available non-text media. While SMMILE currently focuses solely on non-text media in the form of images to easily benchmark various VLMs, this abstract format will enable us to extend to additional modalities in the future.

The problem creation pipeline for SMMILE involves a guided, step-by-step workflow. First, the clinical expert is presented with a set of detailed instructions, which cover topic scope, data sourcing, and answer formatting (Appendix B.1). Next, the expert is directed to the homepage interface, where they initialize a new problem (Appendix B.2). Then, the problem creation tool is loaded (Appendix B.3), which enables the expert to select the relevant medical specialty as well as add, remove, or reorder panels for in-context examples and the final query. Finally, upon clicking "Submit," the expert is shown an overview of the completed problem for validation or further editing

(Appendix B.4). Our pipeline is designed to ensure consistency, adherence to guidelines, and ease of use through the problem creation process.

We then performed manual quality control, which involved the following three steps. First, each problem was manually inspected by two different authors to check for errors, irregularities, or other inaccuracies. Second, each problem was annotated and categorized as shown in Figure 3 (A-F). Third, all text was put through spell check software and manually adjusted when needed. This resulted in 15 grammar and spelling changes to questions and 6 grammar and spelling changes to answers. Additionally, 8 problems had to be modified to make exact match (EM) evaluations possible, including 1 phrasing change, 5 insertions of additional in-context examples, and 2 query edits.

## 2.2 Benchmark Details

The SMMILE dataset includes 111 problems, with each problem consisting of a single query and an average of 3.65 in-context examples (with a spread of 2 to 19 examples per problem). In total, SMMILE encompasses 517 question-image-answer triplets.

Figure 3 analyzes the composition of SMMILE with several descriptive statistics. As shown in Graphs A-C, the dataset is primarily composed of diagnostic and classification problems, with about three-quarters requiring free response formats. While many cases are common in clinical practice, over one-third represent uncommon presentations. Graph D characterizes problem difficulty, as rated by clinical experts during data curation. Specifically, experts considered: (1) whether the problem required complex multimodal reasoning beyond simple visual pattern matching, (2) the degree of specialized medical knowledge needed, and (3) to what extent successful answers would likely require leveraging the provided in-context examples rather than relying solely on pre-trained knowledge. Graphs E and F demonstrate a diversity of image types, covering 6 medical specialties and 13 imaging modalities. Graph G summarizes the distribution of in-context examples per problem, and Graph H details the distribution of question and answer lengths across the dataset.

We leverage the SMMILE dataset to design two benchmark tasks: (1) open-ended generation, where a MLLM is presented with a query question and image and tasked with generating a free-text response, and (2) closed-ended generation, where the MLLM selects an answer from a closed set of possible choices obtained from the in-context example set. Additionally, we introduce a large-scale, augmented dataset called SMMILE++ by permuting in-context examples from a subset of problems in the original dataset. To find permutable problems, we excluded all reasoning problems. An upper limit of $4! = 24$ permutations per problem was used, implying that problems with more than 4 in-context examples were shuffled until 24 unique permutations had been created. The final SMMILE++ dataset includes 1038 problems. Descriptive statistics for SMMILE++ are presented in Appendix Section C.

## 3 Experiments

We now utilize the SMMILE benchmark to evaluate the extent to which existing MLLMs can learn relevant medical knowledge when presented with multimodal in-context examples. We describe our experimental setup in Section 3.1, and we provide quantitative analyses of 15 open-source and closed-source MLLMs in Sections 3.2 and 3.3. Our results show that existing MLLMs struggle to effectively learn from multimodal in-context examples in the medical setting, demonstrating that SMMILE is a challenging and practically-useful benchmark for future MLLM development.

### 3.1 Experimental setup

**Models**  We evaluate a total of 15 state-of-the-art MLLMs encompassing a range of model sizes (0.5B to 90B parameters for the open-source models), pretraining domains (general-purpose MLLMs and domain-specific medical MLLMs), access types (open-source and closed-source), and model architectures. Open-source models considered in this work include: LLaVA-v1.5 (7B and 13B) [22], LLaVA-NeXT-7B [23], LLaVA-OneVision (0.5B and 7B) [19], LLaVA-Med-7B [20], Llama-3.2-Vision-90B [9], MedVLM-R1 [27], MedGemma 4B [8], and Qwen2.5-VL (3B, 7B, 32B, and 72B) [3]. In particular, LLaVA-Med-7B, MedGemma 4B, and MedVLM-R1 are domain-specific MLLMs designed specifically for medical tasks. Closed-source models considered in this work include GPT-4o [26] and Claude 3.7 Sonnet [2]. For all models, we use a standard input prompt consisting of

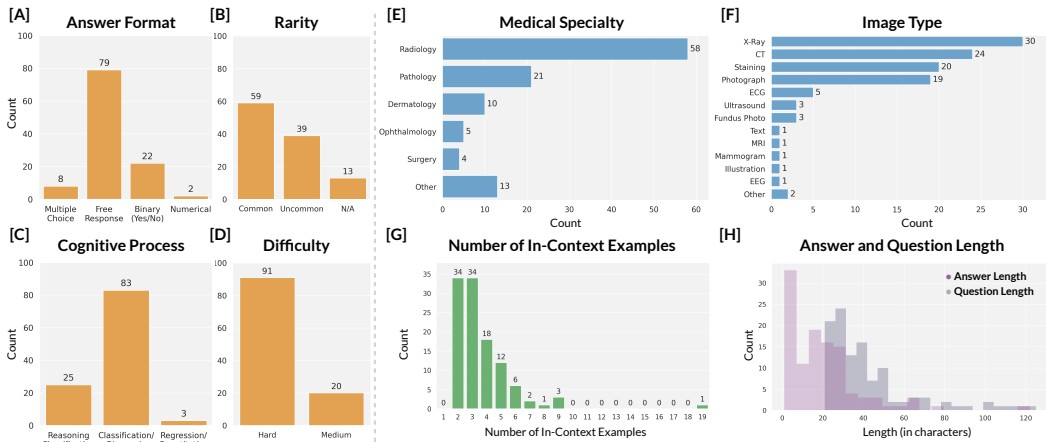

Figure 3: Dataset characteristics. (A–D) Distribution of four key categorical annotations across the unique problems: (A) answer format, (B) rarity of the clinical case based on how often clinicians would experience the medical concepts included in each problem, (C) primary cognitive process required (where reasoning classification is defined by final problem not having direct support in its in-context example set), and (D) rated difficulty. (E–F) Horizontal barplots showing the breakdown of each problem by its main medical specialty (E) and by main image type used (F). (G) Histogram of the number of in-context examples provided per problem. (H) Overlaid histograms of the character-length distributions for questions versus answers. All panels are based on the 111 problems included in SMMILE.

a system message, in-context examples, and the query image and question. To ensure fair comparison, we set the maximum generation length to 512 tokens for all models across all open-ended tasks.

**Baselines** In addition to the 15 MLLMs evaluated above, we consider three baselines: (1) Random, where a random answer from the in-context example set is selected as the response, (2) Majority, where the most frequent answer from the in-context example set is selected as the response, and (3) Text-Only, where a text-only LLM (Llama3.3 70B) [9] is evaluated using only the textual components (questions and answers) from the problems, without any image inputs. For the ICL evaluation, this text-only model receives the questions and answers from the in-context examples, while for the 0-shot evaluation, it receives only the query question.

**Evaluation Metrics** For *open-ended* evaluations, we evaluate MLLM-generated outputs using two metrics. First, the **Exact Match (EM)** metric counts a model generation as correct (score of 100) if it exactly matches the ground-truth answer, and incorrect (score of 0) otherwise. During evaluation, answers are normalized to account for minor variations in formatting, punctuation, and capitalization before comparison. Second, the **LLM-as-a-Judge** approach provides a text-only LLM (Llama3.3 70B) with both the model generation and the ground-truth answer; the model is then prompted to evaluate accuracy. The LLM provides a binary judgment (0 for incorrect, 1 for correct) for each generated output, and the final score represents the percentage of outputs judged as correct. For *closed-ended* (multiple-choice) evaluations, we measure the accuracy of selecting the correct option.

To estimate sampling variability in our metrics, we employ a bootstrap resampling approach with $N_{\text{bootstrap}} = 1000$ bootstrap iterations. For each iteration, we randomly sample with replacement from the original results to create a simulated dataset of the same size as the original dataset, then calculate the accuracy for this bootstrap sample. We report the mean accuracy and standard deviation across all 1000 bootstrap samples. We use a fixed random seed for reproducibility.

To complement automated metrics, we conducted **Human Expert** evaluations, where five clinical experts independently evaluated model responses in both zero-shot and ICL settings. Each response was assessed by two different clinicians using binary ratings (correct/incorrect). Inter-rater agreement was perfect in the ICL setting (100%) and ranged from 98.2% to 100% in the zero-shot setting, demonstrating high reliability.

Additional experimental details are provided in Appendix Section D.

## 3.2 Benchmarking MLLMs with SMMILE

In Table 1, we report performance metrics resulting from evaluations of 15 state-of-the-art MLLMs on SMMILE. Several trends are observable from these results:

- **ICL shows mixed results with concerning baseline failures despite average improvements.** While the average performance improvement across the 15 models in the open-ended LLM-as-a-Judge setting is 8.01% (absolute) and 31.2% (relative), this masks a troubling heterogeneity in ICL effectiveness. Notably, 7 out of 15 models perform worse than even a Random baseline (randomly selecting an answer from in-context examples, 27.86%): MedVLM-R1 (26.74%), LLaVA-OneVision-7B (24.25%), LLaVA-NeXT-7B (23.66%), LLaVA-OneVision-0.5B (21.63%), LLaVA-v1.5-13B (20.91%), LLaVA-v1.5-7B (18.727%), and LLaVA-Med-7B (10.19%). Some models even show performance degradation with ICL, such as LLaVA-Med-7B dropping by more than half from 21.65% to 10.19%. The average improvement is driven primarily by a few models showing substantial gains: LLaVA-NeXT-7B with 107.2% (11.42%→23.66%), Qwen2.5-VL-32B with 65.4% (25.27%→41.79%), and Qwen2.5-VL-72B with 42.4% (29.90%→42.59%) relative improvement, respectively. This highly variable performance reveals that ICL benefits remain model-specific and unreliable.

- **GPT-4o is the overall leader.** With ICL, GPT-4o delivers the best open-ended score (49.88%) and the best closed-ended accuracy (58.85%), demonstrating the most effective multimodal reasoning capabilities across task formats.

- **Domain-specific medical models do not perform significantly better than general-purpose baselines of comparable size (1-10B parameters).** Across evaluations against size-matched general-purpose models, medical models demonstrate highly variable performance. MedGemma 4B achieves similar zero-shot and ICL LLM-as-a-Judge performance when compared to similarly-sized general-purpose models such as Qwen2.5-VL-3B. However, in some instances, ICL leads to *performance drops*; in particular, LLaVA-Med-7B exhibits severe degradation with ICL when evaluated via LLM-as-a-Judge, dropping to a fraction of its zero-shot performance (21.65% to 10.19%). Such trends are not observed for size-matched general-purpose models like LLaVA-NeXT-7B and LLaVA-v1.5-7B. These findings indicate that domain-specific fine-tuning does not consistently improve the ICL capabilities of models when compared to general-purpose models of similar scale.

- **Model scale is not the sole determinant of success.** Smaller Qwen (3-7B) and LLaVA (0.5-7B) variants lag behind larger models. However, the Qwen2.5-VL-32B model approximately matches or outperforms its 72B counterpart.

- **Qwen2.5-VL-32B achieves the highest performance when evaluated with exact match, yet exact match remains challenging.** Qwen2.5-VL-32B achieves the highest EM accuracy (31.84%), followed closely by Llama-3.2-Vision-90B (30.53%). This shows that large-scale models are capable of translating ICL into substantially better literal answer matching. However, EM scores trail far behind closed-ended performance and LLM-as-a-Judge performance, underscoring the difficulty of word-for-word answer generation.

- **Closed-ended questions are easier for MLLMs.** Five models achieve an accuracy greater than 50% on closed-ended evaluations (GPT-4o, Claude 3.7 Sonnet, Llama-3.2-Vision-90B, Qwen2.5-VL-32B, Qwen2.5-VL-72B). A text-only baseline still achieves an accuracy of 38.6%, suggesting that multiple-choice items rely less on precise visual grounding than open-ended generation.

- **Expert evaluation validates automated metrics while revealing some differences.** Human expert ratings show strong correlation with LLM-as-a-Judge scores in the zero-shot setting (Pearson $r = 0.84, p < 0.0001$) but only moderate correlation in the ICL setting ($r = 0.72, p = 0.003$). We find that expert ratings tend to be conservative: medical experts rate some models substantially lower than LLM-as-a-Judge in ICL settings. For example, Qwen2.5-VL-32B and Qwen2.5-VL-72B receive expert ratings of 31-33% despite LLM-as-a-Judge scores of 41-43% in the ICL setting. This suggests that LLM-as-a-Judge may be overly lenient in ICL settings, potentially accepting responses that match the format and phrasing of in-context examples without ensuring clinical adequacy. The high inter-rater agreement among experts ($\geq 98.2\%$) demonstrates that their ratings provide crucial complementary signal to our automated metrics.

Table 1: We benchmark 15 MLLMs on SMMILE, reporting zero-shot performance and performance with in-context examples. The best result is bolded for each task and evaluation metric. *Text-only baseline used Llama 3.3 70B. **Llava-Med-7B refers to LLaVA-Med-v1.5-Mistral-7B. †Zero-shot EM scores were ≤ 3.65% for all models and are omitted.

| Model | Open-ended | | | | | Closed-ended | |
| | LLM-as-a-Judge | | Expert Rating | | EM | MCQA | |
| | 0-shot | ICL | 0-shot | ICL | ICL† | 0-shot | ICL |
|---|---|---|---|---|---|---|---|
| Majority | – | $26.30_{\pm3.88}$ | – | – | $27.26_{\pm4.03}$ | – | $24.15_{\pm3.70}$ |
| Random | – | $27.86_{\pm4.43}$ | – | – | $23.16_{\pm3.88}$ | – | $36.30_{\pm4.66}$ |
| Text only* | $5.32_{\pm2.27}$ | $16.53_{\pm3.94}$ | – | – | $5.22_{\pm1.70}$ | $38.62_{\pm4.61}$ | $28.03_{\pm4.28}$ |
| Claude 3.7 Sonnet | $\mathbf{37.18}_{\pm4.39}$ | $36.17_{\pm4.44}$ | $\mathbf{39.19}$ | $\mathbf{44.14}$ | $2.63_{\pm1.67}$ | $\mathbf{56.10}_{\pm4.31}$ | $42.01_{\pm4.83}$ |
| GPT-4o | $32.56_{\pm4.60}$ | $\mathbf{49.88}_{\pm4.69}$ | $33.33$ | $43.24$ | $8.94_{\pm2.54}$ | $49.74_{\pm4.48}$ | $\mathbf{58.85}_{\pm4.62}$ |
| Llama-3.2-Vision-90B | $31.84_{\pm4.38}$ | $40.66_{\pm4.99}$ | $36.94$ | $33.33$ | $\mathbf{30.53}_{\pm4.07}$ | $55.04_{\pm4.93}$ | $30.30_{\pm5.20}$ |
| LLaVA-v1.5-7B | $14.61_{\pm3.57}$ | $18.72_{\pm3.40}$ | $17.12$ | $21.62$ | $16.37_{\pm3.32}$ | $40.34_{\pm5.35}$ | $22.30_{\pm3.92}$ |
| LLaVA-v1.5-13B | $19.58_{\pm3.64}$ | $20.91_{\pm3.49}$ | $22.52$ | $26.13$ | $19.54_{\pm3.95}$ | $38.96_{\pm4.83}$ | $24.92_{\pm4.25}$ |
| LLaVA-NeXT-7B | $11.42_{\pm3.04}$ | $23.66_{\pm3.90}$ | $13.51$ | $33.33$ | $2.69_{\pm1.31}$ | $38.11_{\pm4.26}$ | $29.01_{\pm4.05}$ |
| LLaVA-OneVision-0.5B | $18.26_{\pm3.93}$ | $21.63_{\pm4.00}$ | $19.82$ | $18.02$ | $13.46_{\pm3.11}$ | $44.03_{\pm4.47}$ | $32.11_{\pm4.44}$ |
| LLaVA-OneVision-7B | $16.41_{\pm3.65}$ | $24.25_{\pm3.81}$ | $25.23$ | $27.03$ | $22.43_{\pm4.33}$ | $40.15_{\pm4.59}$ | $27.17_{\pm4.28}$ |
| LLaVA-Med-7B** | $21.65_{\pm4.18}$ | $10.19_{\pm3.06}$ | $22.52$ | $10.81$ | $0.00_{\pm0.00}$ | $0.00_{\pm0.00}$ | $0.00_{\pm0.00}$ |
| MedGemma-4B-Multimodal | $27.73_{\pm4.70}$ | $36.86_{\pm4.81}$ | $27.03$ | $32.43$ | $12.14_{\pm3.07}$ | $41.21_{\pm5.15}$ | $40.67_{\pm4.93}$ |
| MedVLM-R1 | $25.20_{\pm4.09}$ | $26.74_{\pm4.44}$ | $25.23$ | $24.32$ | $15.26_{\pm2.91}$ | $36.54_{\pm4.63}$ | $26.22_{\pm4.31}$ |
| Qwen2.5-VL-3B | $27.62_{\pm4.26}$ | $33.58_{\pm4.09}$ | $23.42$ | $23.42$ | $26.18_{\pm4.52}$ | $37.58_{\pm4.41}$ | $27.35_{\pm4.23}$ |
| Qwen2.5-VL-7B | $17.90_{\pm3.20}$ | $29.58_{\pm4.63}$ | $11.71$ | $27.03$ | $22.45_{\pm3.81}$ | $38.24_{\pm4.75}$ | $45.01_{\pm4.39}$ |
| Qwen2.5-VL-32B | $25.27_{\pm3.86}$ | $41.79_{\pm4.73}$ | $21.62$ | $32.43$ | $31.84_{\pm4.37}$ | $51.76_{\pm4.46}$ | $49.97_{\pm5.07}$ |
| Qwen2.5-VL-72B | $29.90_{\pm4.08}$ | $42.59_{\pm4.55}$ | $26.13$ | $31.52$ | $15.71_{\pm3.30}$ | $52.33_{\pm4.58}$ | $54.71_{\pm4.89}$ |

In Table 2, we report performance metrics from evaluations of 14 state-of-the-art MLLMs on SMMILE++, the augmented variant of the SMMILE dataset consisting of 1038 problems[1].

- **There are notable changes in model rankings.** Unlike Table 1 where GPT-4o was dominant, Qwen2.5-VL-72B now takes the lead (53.8% LLM-as-a-Judge accuracy, 63.2% on ICL-MCQA).

- **Broader benefits from in-context learning are visible.** Larger relative performance improvements are observed when in-context examples are presented to models, such as: LLaVA-v1.5-7B (99.4% relative improvement in LLM-as-a-Judge, 12.31% → 24.55%), Qwen2.5-VL-7B (94.4% relative improvement in LLM-as-a-Judge, 21.10% → 41.01%), and Qwen2.5-VL-3B (79.9% relative improvement in LLM-as-a-Judge, 17.53% → 31.54%). In the open-ended setting, all models (with the notable exception of LLaVA-Med-7B) demonstrate higher ICL performance than zero-shot performance when evaluated with LLM-as-a-Judge, with an average relative improvement of 44.7%.

- **Exact-match is still challenging, but the ceiling rises.** The best ICL accuracy with EM evaluation increases from 31.84% (Qwen2.5-VL-32B in Table 1) to 35.34% (Qwen2.5-VL-7B in Table 2).

- **Closed-ended tasks remain easier.** Top MCQA performance climbs from 58.9% (GPT-4o in Table 1) to 63.2% (Qwen2.5-VL-72B in Table 2), and four models (GPT-4o, Qwen2.5-VL-7B, Qwen2.5-VL-32B, and Qwen2.5-VL-72B) surpass the 50% mark. These results are consistent with the trend that multiple-choice questions are less challenging than open-ended generation.

### 3.3 Fine-Grained Analysis

We now perform a fine-grained breakdown of MLLM performance across the SMMILE benchmark. We specifically focus on five reproducible MLLMs for this analysis: LLaVA-OneVision-0.5B, LLaVA-Med-7B, LLaVA-v1.5-13B, Qwen2.5-VL-32B, and Qwen2.5-VL-72B. We first evaluate MLLM performance stratified by answer format. Each ground-truth answer in the SMMILE dataset was annotated by an expert with one of the following four categorical labels: "multiple choice", "free response", "binary (yes/no)", or "numerical". As shown in Figure 4 (Panel A), MLLMs display substantial variations in performance across the four categories, with all five evaluated models demonstrating the strongest performance on binary (yes/no) answers. Notably, we find that all evaluated models fail to correctly answer questions with numerical answers, which is a critical limitation since the ability to provide quantitative responses is vital for effective decision-making in

---

[1]Claude 3.7 Sonnet was excluded from evaluations on SMMILE++ due to limited access and API usage fees.

Table 2: We benchmark 14 state-of-the-art MLLMs on SMMILE++, the augmented variant of the SMMILE dataset with 1038 samples. We report both zero-shot performance as well as performance with in-context examples. The best result is bolded for each task and evaluation metric. *Text-only baseline used Llama 3.3 70B. **Llava-Med-7B refers to LLaVA-Med-v1.5-Mistral-7B.

| Model | Open-ended | | | | Closed-ended | |
| | LLM-as-a-Judge | | EM | | MCQA | |
| | 0-shot | ICL | 0-shot | ICL | 0-shot | ICL |
|---|---|---|---|---|---|---|
| Majority | - | $17.70 \pm_{1.19}$ | - | $17.59 \pm_{1.19}$ | - | $16.65 \pm_{1.13}$ |
| Random | - | $25.35 \pm_{1.34}$ | - | $25.35 \pm_{1.32}$ | - | $33.77 \pm_{1.46}$ |
| Text only* | $7.37 \pm_{0.84}$ | $14.55 \pm_{1.10}$ | $0.00 \pm_{0.00}$ | $3.59 \pm_{0.58}$ | $41.45 \pm_{1.52}$ | $22.41 \pm_{1.27}$ |
| GPT-4o | $\mathbf{38.41} \pm_{1.51}$ | $46.45 \pm_{1.54}$ | $0.00 \pm_{0.00}$ | $7.79 \pm_{0.81}$ | $56.70 \pm_{1.48}$ | $55.76 \pm_{1.56}$ |
| LLama-3.2-Vision-90B | $25.23 \pm_{1.38}$ | $29.56 \pm_{1.43}$ | $0.00 \pm_{0.00}$ | $27.51 \pm_{1.38}$ | $49.27 \pm_{1.50}$ | $30.04 \pm_{1.40}$ |
| LLaVA-v1.5-7B | $12.31 \pm_{1.07}$ | $24.55 \pm_{1.35}$ | $0.00 \pm_{0.00}$ | $20.47 \pm_{1.22}$ | $48.32 \pm_{1.55}$ | $23.83 \pm_{1.34}$ |
| LLaVA-v1.5-13B | $14.23 \pm_{1.07}$ | $23.13 \pm_{1.30}$ | $0.00 \pm_{0.00}$ | $21.12 \pm_{1.28}$ | $41.33 \pm_{1.51}$ | $25.55 \pm_{1.27}$ |
| LLaVa-NeXT-7B | $16.39 \pm_{1.14}$ | $17.57 \pm_{1.16}$ | $0.00 \pm_{0.00}$ | $3.53 \pm_{0.55}$ | $42.26 \pm_{1.46}$ | $26.15 \pm_{1.35}$ |
| LLaVA-OneVision-0.5B | $17.75 \pm_{1.14}$ | $20.16 \pm_{1.22}$ | $\mathbf{6.92} \pm_{0.78}$ | $14.28 \pm_{1.09}$ | $35.51 \pm_{1.43}$ | $27.78 \pm_{1.41}$ |
| LLaVA-OneVision-7B | $20.41 \pm_{1.25}$ | $27.72 \pm_{1.37}$ | $2.91 \pm_{0.54}$ | $25.70 \pm_{1.31}$ | $41.64 \pm_{1.47}$ | $27.45 \pm_{1.37}$ |
| LLaVA-Med-7B** | $24.84 \pm_{1.31}$ | $4.62 \pm_{0.64}$ | $0.00 \pm_{0.00}$ | $0.19 \pm_{0.14}$ | $0.29 \pm_{0.17}$ | $0.00 \pm_{0.00}$ |
| MedGemma-4B-Multimodal | $24.72 \pm_{1.35}$ | $38.66 \pm_{1.54}$ | $0.00 \pm_{0.00}$ | $13.30 \pm_{1.06}$ | $40.36 \pm_{1.55}$ | $44.78 \pm_{1.52}$ |
| MedVLM-R1 | $28.82 \pm_{1.37}$ | $33.54 \pm_{1.46}$ | $2.91 \pm_{0.54}$ | $24.32 \pm_{1.33}$ | $37.68 \pm_{1.54}$ | $24.09 \pm_{1.34}$ |
| Qwen2.5-VL-3B | $17.53 \pm_{1.22}$ | $31.54 \pm_{1.49}$ | $0.00 \pm_{0.00}$ | $28.09 \pm_{1.38}$ | $41.72 \pm_{1.50}$ | $38.10 \pm_{1.55}$ |
| Qwen2.5-VL-7B | $21.10 \pm_{1.24}$ | $41.01 \pm_{1.56}$ | $0.00 \pm_{0.00}$ | $\mathbf{35.34} \pm_{1.45}$ | $54.38 \pm_{1.55}$ | $49.79 \pm_{1.60}$ |
| Qwen2.5-VL-32B | $28.92 \pm_{1.40}$ | $35.37 \pm_{1.48}$ | $0.00 \pm_{0.00}$ | $25.26 \pm_{1.33}$ | $46.56 \pm_{1.57}$ | $53.25 \pm_{1.53}$ |
| Qwen2.5-VL-72B | $34.79 \pm_{1.44}$ | $\mathbf{53.80} \pm_{1.54}$ | $0.00 \pm_{0.00}$ | $24.44 \pm_{1.33}$ | $\mathbf{60.62} \pm_{1.55}$ | $\mathbf{63.22} \pm_{1.51}$ |

medical settings. This finding is corroborated by analysis in Figure 4 (Panel B), which again finds that all evaluated MLLMs struggle when quantitative reasoning is necessary to answer a question.

In Figure 4 (Panel C), we report the effect of the number of ICL examples on MLLM performance. For all evaluated models, providing two ICL examples leads to substantial improvements in performance over the zero-shot setting. However, trends become more variable as the number of ICL examples increases. In particular, we observe that increasing the number of ICL examples is *not* consistently correlated with stronger performance; in particular, all models exhibit substantial performance degradations that often dip below zero-shot performance. These results suggest that existing MLLMs may be unable to perform ICL tasks when provided with lengthy inputs consisting of multiple interleaved image-text pairs.

Figure 4 (Panel D) shows that MLLMs exhibit highly variable performance across the 13 included imaging modalities. No MLLM achieves strong performance across all modalities, suggesting that the MLLMs are unable to consistently glean relevant information from provided in-context examples. When query images come from the MRI and illustration modalities, all evaluated models fail to generate any correct answers. The text, mammogram, fundus photograph, and EEG modalities also prove to be challenging, with at least two MLLMs failing to generate any correct answers.

In summary, our results demonstrate that SMMILE is a comprehensive and challenging benchmark for evaluating in-context learning abilities of MLLMs in medical settings. We hope that SMMILE can serve as a valuable resource for driving forward the development of future MLLMs. Extended fine-grained analyses are provided in Appendix Sections E and F.

## 4 Analyzing In-Context Example Construction

The manually-curated and high-quality nature of the SMMILE benchmark can help reveal insights into how effective in-context examples can be selected for MLLMs. In this section, we analyze two critical factors associated with selecting in-context examples: (1) quality of in-context examples (Section 4.1) and (2) the order of examples provided to the MLLM (Section 4.2).

### 4.1 Analyzing Example Quality

The role of in-context example quality on MLLM performance is not well-understood, and the high-quality nature of SMMILE provides a unique opportunity for addressing this question. Here, we create two perturbed versions of the SMMILE dataset as follows. (1) *SMMILE-Random-Noise*: For

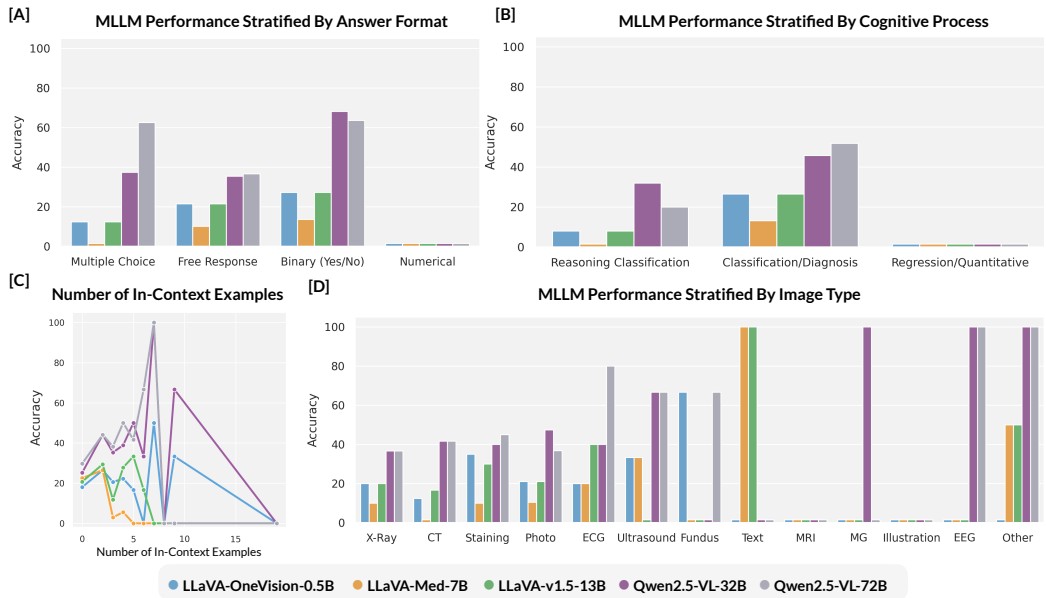

Figure 4: We provide a fine-grained breakdown of MLLM performance on the SMMILE benchmark. We report performance stratified by answer format (Panel A), cognitive process necessary to obtain the answer (Panel B), number of in-context examples provided to the model (Panel C), and image type (Panel D). Here, we focus on open-ended evaluations, and the y-axis represents prediction accuracy as computed by the LLM-as-a-Judge approach. The acronym MG refers to Mammograms.

each sample in SMMILE, we insert a random image-question-answer triplet from the dataset to the in-context example set. (2) *SMMILE-Targeted-Noise*: For each sample in SMMILE, we insert an image-question-answer triplet from the dataset that shares the same specialty as the sample.

In Table 3, we report performance of 9 MLLMs across these perturbed variants of SMMILE. We observe that the inclusion of just one noisy sample in the in-context example list can impair performance, with most models exhibiting performance degradations on both SMMILE-Random-Noise (9.1% relative decrease from SMMILE on average) and SMMILE-Targeted-Noise (9.5% relative decrease from SMMILE on average). Targeted noise contributes to slightly lower performance than random noise on average, suggesting that even targeted, specialty-based selection of in-context examples can impair performance if the selected examples are not effective demonstrations of the task at hand. Importantly, the effects of noise are *model-specific*; the presence of noisy in-context examples affects each model in differing ways, leading to substantial changes in the final rankings. Our results demonstrate the critical need for high-quality, manually-curated benchmarks for evaluating in-context abilities of MLLMs in the medical setting, as the presence of noisy or irrelevant samples in the in-context example set can prevent developers from accurately understanding model capabilities.

## 4.2 Analyzing Example Order

Prior works have suggested that models may be sensitive to the order of in-context examples [36, 10, 31]. Here, we investigate the extent to which (a) the first in-context example and (b) the last in-context example influence MLLM predictions. To this end, we filter the SMMILE dataset to a subset of 69 problems where at least one in-context example has an identical answer to the query question; then, we modify the ordering of the in-context example list such that the placement of examples with identical answers can be explicitly controlled.

In Figure 5 (left), we compare performance when the *first* in-context example contains an identical answer to the query question ("Yes") with performance when examples with identical answers occur later in the in-context example list ("No"). We observe substantial performance degradations (absolute decrease of up to 47%) when the answer to the first in-context example matches the answer to the query question. This trend holds for all nine MLLMs evaluated in this setting, which consist of varied architectures and parameter counts. Importantly, our finding suggests that MLLMs are

Table 3: We create two perturbed versions of the SMMILE dataset (SMMILE-Random-Noise and SMMILE-Targeted-Noise) in order to evaluate the role of in-context example quality on MLLM performance. Here, we report performance across nine open-source MLLMs (ordered by model size) in the open-ended setting with LLM-as-a-Judge evaluation. The best result per row is bolded.

| Model | SMMILE | SMMILE-Random-Noise | SMMILE-Targeted-Noise |
|---|---|---|---|
| LLaVA-OneVision-0.5B | **21.63** $\pm_{4.00}$ | 19.41 $\pm_{3.50}$ | 21.35 $\pm_{3.83}$ |
| Qwen2.5-VL-3B | **33.58** $\pm_{4.09}$ | 30.40 $\pm_{4.65}$ | 30.37 $\pm_{4.73}$ |
| LLaVA-v1.5-7B | **18.72** $\pm_{3.40}$ | 17.95 $\pm_{3.87}$ | 14.80 $\pm_{3.31}$ |
| LLaVA-OneVision-7B | **24.25** $\pm_{3.81}$ | 21.90 $\pm_{3.86}$ | 23.04 $\pm_{3.82}$ |
| LLaVA-NeXT-7B | 23.66 $\pm_{3.90}$ | 17.77 $\pm_{3.26}$ | **24.38** $\pm_{3.99}$ |
| LLaVA-Med-7B | **10.19** $\pm_{3.06}$ | 4.88 $\pm_{2.07}$ | 1.88 $\pm_{1.32}$ |
| Qwen2.5-VL-7B | 29.58 $\pm_{4.63}$ | **33.11** $\pm_{3.92}$ | 31.92 $\pm_{4.01}$ |
| LLaVA-v1.5-13B | **20.91** $\pm_{3.49}$ | 18.87 $\pm_{3.80}$ | 16.14 $\pm_{3.40}$ |
| Qwen2.5-VL-32B | **41.79** $\pm_{4.73}$ | 39.60 $\pm_{4.60}$ | 39.10 $\pm_{4.48}$ |
| Average | **24.92** | 22.65 | 22.55 |

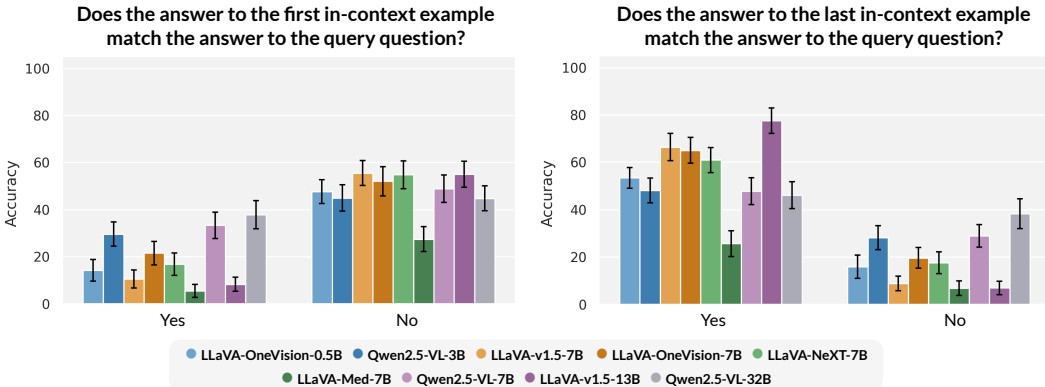

Figure 5: We analyze the effect of example order on MLLM performance. We report performance across 9 MLLMs (ordered by model size) in the open-ended setting with LLM-as-a-Judge evaluation.

affected by **recency bias**, where placing the most relevant in-context examples (i.e. those that share answers with query question) later in the list can lead to improved performance. This finding is further corroborated by results in Figure 5 (right), where we compare performance when the *last* in-context example contains an identical answer to the query question ("Yes") with performance when examples with identical answers occur earlier in the in-context example list ("No"). We observe substantial performance improvements (absolute improvement of up to 71%) when the answer to the last in-context example matches the answer to the query question.

## 5 Discussion

**Key findings.** In this work, we introduced SMMILE, a multimodal medical in-context learning benchmark designed in collaboration with a team of international clinical experts. Even the best-performing models, such as GPT-4o on SMMILE and Qwen2.5-VL-72B on SMMILE++, are only capable of answering approximately half of the questions accurately. Applying ICL results in substantial performance boosts for only a few models. Our results demonstrate a significant gap between current MLLMs and the generalizability required for clinical use. Limitations and future work are discussed in Appendix Section G.

**Impact.** SMMILE is the *first* benchmark to (i) evaluate multimodal in-context learning in medicine, (ii) release expert-annotated problems with graded task difficulty for supporting medical ICL, and (iii) supply a fine-grained analysis toolkit with open datasets, evaluation code, and baselines so that researchers can reproduce our pipeline and measure progress with minimal friction.

## Acknowledgments

MV is supported by graduate fellowship awards from the Department of Defense (NDSEG), the Knight-Hennessy Scholars program at Stanford University, and the Quad program. JBD was supported in part by the Medical Imaging and Data Resource Center (MIDRC), funded by the National Institute of Biomedical Imaging and Bioengineering (NIBIB) of the National Institutes of Health under contract 75N92020D00021 and through The Advanced Research Projects Agency for Health (ARPA-H). We thank Dr. Tina Chen (Stanford Radiology Resident) for her contribution to the SMMILE dataset curation.

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

# NeurIPS Paper Checklist

1. **Claims**

   Question: Do the main claims made in the abstract and introduction accurately reflect the paper's contributions and scope?

   Answer: [Yes]

   Justification: The abstract and introduction clearly state the goals of our paper: we present SMMILE, the first benchmark for multimodal medical in-context learning, test 14 state-of-the-art models, and show that most models gain little from in-context learning while suffering from recency bias and noise. These claims match the results in the tables and figures, and the text makes it clear that SMMILE is only a step toward stronger clinical AI, not a complete fix.

   Guidelines:

   - The answer NA means that the abstract and introduction do not include the claims made in the paper.
   - The abstract and/or introduction should clearly state the claims made, including the contributions made in the paper and important assumptions and limitations. A No or NA answer to this question will not be perceived well by the reviewers.
   - The claims made should match theoretical and experimental results, and reflect how much the results can be expected to generalize to other settings.
   - It is fine to include aspirational goals as motivation as long as it is clear that these goals are not attained by the paper.

2. **Limitations**

   Question: Does the paper discuss the limitations of the work performed by the authors?

   Answer: [Yes]

   Justification: The paper discusses broader impacts in Section 5 and limitations and future work in Appendix Section G. The paper acknowledges limitations related to dataset scalability, limited modality coverage, and potential biases in expertise distribution among contributors. The paper also discusses how these limitations could be addressed in future work through expanded contributor networks, increased dataset size, and inclusion of additional medical modalities beyond the current focus on images.

   Guidelines:

   - The answer NA means that the paper has no limitation while the answer No means that the paper has limitations, but those are not discussed in the paper.
   - The authors are encouraged to create a separate "Limitations" section in their paper.
   - The paper should point out any strong assumptions and how robust the results are to violations of these assumptions (e.g., independence assumptions, noiseless settings, model well-specification, asymptotic approximations only holding locally). The authors should reflect on how these assumptions might be violated in practice and what the implications would be.
   - The authors should reflect on the scope of the claims made, e.g., if the approach was only tested on a few datasets or with a few runs. In general, empirical results often depend on implicit assumptions, which should be articulated.
   - The authors should reflect on the factors that influence the performance of the approach. For example, a facial recognition algorithm may perform poorly when image resolution is low or images are taken in low lighting. Or a speech-to-text system might not be used reliably to provide closed captions for online lectures because it fails to handle technical jargon.
   - The authors should discuss the computational efficiency of the proposed algorithms and how they scale with dataset size.
   - If applicable, the authors should discuss possible limitations of their approach to address problems of privacy and fairness.

- While the authors might fear that complete honesty about limitations might be used by reviewers as grounds for rejection, a worse outcome might be that reviewers discover limitations that aren't acknowledged in the paper. The authors should use their best judgment and recognize that individual actions in favor of transparency play an important role in developing norms that preserve the integrity of the community. Reviewers will be specifically instructed to not penalize honesty concerning limitations.

3. **Theory assumptions and proofs**

   Question: For each theoretical result, does the paper provide the full set of assumptions and a complete (and correct) proof?

   Answer: [NA]

   Justification: This paper does not include theoretical results.

   Guidelines:

   - The answer NA means that the paper does not include theoretical results.
   - All the theorems, formulas, and proofs in the paper should be numbered and cross-referenced.
   - All assumptions should be clearly stated or referenced in the statement of any theorems.
   - The proofs can either appear in the main paper or the supplemental material, but if they appear in the supplemental material, the authors are encouraged to provide a short proof sketch to provide intuition.
   - Inversely, any informal proof provided in the core of the paper should be complemented by formal proofs provided in appendix or supplemental material.
   - Theorems and Lemmas that the proof relies upon should be properly referenced.

4. **Experimental result reproducibility**

   Question: Does the paper fully disclose all the information needed to reproduce the main experimental results of the paper to the extent that it affects the main claims and/or conclusions of the paper (regardless of whether the code and data are provided or not)?

   Answer: [Yes]

   Justification: Our paper provides comprehensive information to reproduce the main experimental results that support our claims and conclusions. We include detailed descriptions of our benchmarking methodology, evaluation metrics, and experimental setup. Our code is publicly available. The majority of models evaluated in this work are open-source and can be accessed through the repositories referenced in our paper, while two require API keys for access. We specify all hyperparameters, preprocessing steps, and evaluation protocols to ensure full reproducibility. Our codebase also includes information about computational resources required and any additional tools or libraries necessary for implementation, with all dependencies documented.

   Guidelines:

   - The answer NA means that the paper does not include experiments.
   - If the paper includes experiments, a No answer to this question will not be perceived well by the reviewers: Making the paper reproducible is important, regardless of whether the code and data are provided or not.
   - If the contribution is a dataset and/or model, the authors should describe the steps taken to make their results reproducible or verifiable.
   - Depending on the contribution, reproducibility can be accomplished in various ways. For example, if the contribution is a novel architecture, describing the architecture fully might suffice, or if the contribution is a specific model and empirical evaluation, it may be necessary to either make it possible for others to replicate the model with the same dataset, or provide access to the model. In general. releasing code and data is often one good way to accomplish this, but reproducibility can also be provided via detailed instructions for how to replicate the results, access to a hosted model (e.g., in the case of a large language model), releasing of a model checkpoint, or other means that are appropriate to the research performed.

- While NeurIPS does not require releasing code, the conference does require all submissions to provide some reasonable avenue for reproducibility, which may depend on the nature of the contribution. For example
    (a) If the contribution is primarily a new algorithm, the paper should make it clear how to reproduce that algorithm.
    (b) If the contribution is primarily a new model architecture, the paper should describe the architecture clearly and fully.
    (c) If the contribution is a new model (e.g., a large language model), then there should either be a way to access this model for reproducing the results or a way to reproduce the model (e.g., with an open-source dataset or instructions for how to construct the dataset).
    (d) We recognize that reproducibility may be tricky in some cases, in which case authors are welcome to describe the particular way they provide for reproducibility. In the case of closed-source models, it may be that access to the model is limited in some way (e.g., to registered users), but it should be possible for other researchers to have some path to reproducing or verifying the results.

5. **Open access to data and code**

Question: Does the paper provide open access to the data and code, with sufficient instructions to faithfully reproduce the main experimental results, as described in supplemental material?

Answer: [Yes]

Justification: Our dataset is openly available on HuggingFace. Our code is available through on GitHub.

Guidelines:

- The answer NA means that paper does not include experiments requiring code.
- Please see the NeurIPS code and data submission guidelines (`https://nips.cc/public/guides/CodeSubmissionPolicy`) for more details.
- While we encourage the release of code and data, we understand that this might not be possible, so "No" is an acceptable answer. Papers cannot be rejected simply for not including code, unless this is central to the contribution (e.g., for a new open-source benchmark).
- The instructions should contain the exact command and environment needed to run to reproduce the results. See the NeurIPS code and data submission guidelines (`https://nips.cc/public/guides/CodeSubmissionPolicy`) for more details.
- The authors should provide instructions on data access and preparation, including how to access the raw data, preprocessed data, intermediate data, and generated data, etc.
- The authors should provide scripts to reproduce all experimental results for the new proposed method and baselines. If only a subset of experiments are reproducible, they should state which ones are omitted from the script and why.
- At submission time, to preserve anonymity, the authors should release anonymized versions (if applicable).
- Providing as much information as possible in supplemental material (appended to the paper) is recommended, but including URLs to data and code is permitted.

6. **Experimental setting/details**

Question: Does the paper specify all the training and test details (e.g., data splits, hyperparameters, how they were chosen, type of optimizer, etc.) necessary to understand the results?

Answer: [Yes]

Justification: Although this is a benchmarking paper that evaluates pre-trained models rather than training new ones, our code clearly specifies all necessary inference hyperparameters such as maximum output tokens and generation configuration details. The evaluation methodology is thoroughly documented, including data processing steps, zero-shot vs. in-context learning modes, multiple choice vs. open-ended formats, and statistical analysis procedures with bootstrap sampling for confidence intervals. The code provides complete

implementation details required to reproduce the benchmarking results, including model loading parameters, tokenization settings, and prompt templates used for inference.

Guidelines:

- The answer NA means that the paper does not include experiments.
- The experimental setting should be presented in the core of the paper to a level of detail that is necessary to appreciate the results and make sense of them.
- The full details can be provided either with the code, in appendix, or as supplemental material.

7. **Experiment statistical significance**

Question: Does the paper report error bars suitably and correctly defined or other appropriate information about the statistical significance of the experiments?

Answer: [Yes]

Justification: The paper reports statistical significance appropriately by including bootstrap-computed confidence intervals alongside all performance metrics. For each evaluated model and condition, accuracy values are presented with corresponding standard deviations (e.g., "$29.01 \pm 4.05$"), clearly indicating the uncertainty in measurements. The methodology section explains that these confidence intervals were derived using bootstrap sampling with 42 as the random seed for reproducibility.

Guidelines:

- The answer NA means that the paper does not include experiments.
- The authors should answer "Yes" if the results are accompanied by error bars, confidence intervals, or statistical significance tests, at least for the experiments that support the main claims of the paper.
- The factors of variability that the error bars are capturing should be clearly stated (for example, train/test split, initialization, random drawing of some parameter, or overall run with given experimental conditions).
- The method for calculating the error bars should be explained (closed form formula, call to a library function, bootstrap, etc.)
- The assumptions made should be given (e.g., Normally distributed errors).
- It should be clear whether the error bar is the standard deviation or the standard error of the mean.
- It is OK to report 1-sigma error bars, but one should state it. The authors should preferably report a 2-sigma error bar than state that they have a 96% CI, if the hypothesis of Normality of errors is not verified.
- For asymmetric distributions, the authors should be careful not to show in tables or figures symmetric error bars that would yield results that are out of range (e.g. negative error rates).
- If error bars are reported in tables or plots, The authors should explain in the text how they were calculated and reference the corresponding figures or tables in the text.

8. **Experiments compute resources**

Question: For each experiment, does the paper provide sufficient information on the computer resources (type of compute workers, memory, time of execution) needed to reproduce the experiments?

Answer: [Yes]

Justification: We provide a detailed breakdown of computational resources in Appendix Section D.

Guidelines:

- The answer NA means that the paper does not include experiments.
- The paper should indicate the type of compute workers CPU or GPU, internal cluster, or cloud provider, including relevant memory and storage.
- The paper should provide the amount of compute required for each of the individual experimental runs as well as estimate the total compute.

- The paper should disclose whether the full research project required more compute than the experiments reported in the paper (e.g., preliminary or failed experiments that didn't make it into the paper).

9. **Code of ethics**

   Question: Does the research conducted in the paper conform, in every respect, with the NeurIPS Code of Ethics https://neurips.cc/public/EthicsGuidelines?

   Answer: [Yes]

   Justification: The paper follows NeurIPS ethics guidelines by: 1) creating a medical benchmark collaboratively with 11 clinical experts, 2) ensuring proper data handling with public URLs referencing openly accessible image content, 3) avoiding patient identifiable information, 4) acknowledging dataset limitations, 5) providing transparent evaluation metrics with proper statistical analysis, and 6) making code and data available to ensure reproducibility. The benchmarking of existing models for medical in-context learning supports ethical advancement in healthcare AI without introducing harmful applications.

   Guidelines:

   - The answer NA means that the authors have not reviewed the NeurIPS Code of Ethics.
   - If the authors answer No, they should explain the special circumstances that require a deviation from the Code of Ethics.
   - The authors should make sure to preserve anonymity (e.g., if there is a special consideration due to laws or regulations in their jurisdiction).

10. **Broader impacts**

    Question: Does the paper discuss both potential positive societal impacts and negative societal impacts of the work performed?

    Answer: [NA]

    Justification: The study is purely experimental: It introduces a benchmark and reports offline model performance without deploying any system in clinical practice or affecting patient care. Because the work is confined to controlled research settings and does not translate into an operational tool, it carries no immediate societal impact to discuss.

    Guidelines:

    - The answer NA means that there is no societal impact of the work performed.
    - If the authors answer NA or No, they should explain why their work has no societal impact or why the paper does not address societal impact.
    - Examples of negative societal impacts include potential malicious or unintended uses (e.g., disinformation, generating fake profiles, surveillance), fairness considerations (e.g., deployment of technologies that could make decisions that unfairly impact specific groups), privacy considerations, and security considerations.
    - The conference expects that many papers will be foundational research and not tied to particular applications, let alone deployments. However, if there is a direct path to any negative applications, the authors should point it out. For example, it is legitimate to point out that an improvement in the quality of generative models could be used to generate deepfakes for disinformation. On the other hand, it is not needed to point out that a generic algorithm for optimizing neural networks could enable people to train models that generate Deepfakes faster.
    - The authors should consider possible harms that could arise when the technology is being used as intended and functioning correctly, harms that could arise when the technology is being used as intended but gives incorrect results, and harms following from (intentional or unintentional) misuse of the technology.
    - If there are negative societal impacts, the authors could also discuss possible mitigation strategies (e.g., gated release of models, providing defenses in addition to attacks, mechanisms for monitoring misuse, mechanisms to monitor how a system learns from feedback over time, improving the efficiency and accessibility of ML).

11. **Safeguards**

Question: Does the paper describe safeguards that have been put in place for responsible release of data or models that have a high risk for misuse (e.g., pretrained language models, image generators, or scraped datasets)?

Answer: [NA]

Justification: The paper focuses on benchmarking existing vision-language models rather than releasing new models with potential misuse risks. The dataset consists of expert-curated medical questions with links to publicly available medical images, which poses minimal safety risks. The paper does not release scraped datasets or pretrained models that would require special safeguards against misuse.

Guidelines:

- The answer NA means that the paper poses no such risks.
- Released models that have a high risk for misuse or dual-use should be released with necessary safeguards to allow for controlled use of the model, for example by requiring that users adhere to usage guidelines or restrictions to access the model or implementing safety filters.
- Datasets that have been scraped from the Internet could pose safety risks. The authors should describe how they avoided releasing unsafe images.
- We recognize that providing effective safeguards is challenging, and many papers do not require this, but we encourage authors to take this into account and make a best faith effort.

12. **Licenses for existing assets**

Question: Are the creators or original owners of assets (e.g., code, data, models), used in the paper, properly credited and are the license and terms of use explicitly mentioned and properly respected?

Answer: [Yes]

Justification: Our research properly credits the original creators of the models used in benchmarking by citing their corresponding papers. In Appendix Section H (Licensing Considerations), our paper explicitly states that the benchmark's question-answer pairs are licensed. The benchmark references medical images via public URLs; the use of these images is subject to the terms and conditions of the respective hosting websites.

Guidelines:

- The answer NA means that the paper does not use existing assets.
- The authors should cite the original paper that produced the code package or dataset.
- The authors should state which version of the asset is used and, if possible, include a URL.
- The name of the license (e.g., CC-BY 4.0) should be included for each asset.
- For scraped data from a particular source (e.g., website), the copyright and terms of service of that source should be provided.
- If assets are released, the license, copyright information, and terms of use in the package should be provided. For popular datasets, paperswithcode.com/datasets has curated licenses for some datasets. Their licensing guide can help determine the license of a dataset.
- For existing datasets that are re-packaged, both the original license and the license of the derived asset (if it has changed) should be provided.
- If this information is not available online, the authors are encouraged to reach out to the asset's creators.

13. **New assets**

Question: Are new assets introduced in the paper well documented and is the documentation provided alongside the assets?

Answer: [Yes]

Justification: The paper introduces a new benchmark dataset (SMMILE) that is thoroughly documented throughout the paper. Section 2 describes the dataset creation process, quality

control measures, and detailed statistics about the benchmark. The benchmark is made available through HuggingFace with accompanying documentation that includes information about licensing, limitations, and usage instructions.

Guidelines:

- The answer NA means that the paper does not release new assets.
- Researchers should communicate the details of the dataset/code/model as part of their submissions via structured templates. This includes details about training, license, limitations, etc.
- The paper should discuss whether and how consent was obtained from people whose asset is used.
- At submission time, remember to anonymize your assets (if applicable). You can either create an anonymized URL or include an anonymized zip file.

14. **Crowdsourcing and research with human subjects**

Question: For crowdsourcing experiments and research with human subjects, does the paper include the full text of instructions given to participants and screenshots, if applicable, as well as details about compensation (if any)?

Answer: [Yes]

Justification: The paper involves clinical experts who created the benchmark dataset and provides details about the instructions given to these experts. The paper includes screenshots of the web interface used for data collection and describes the step-by-step workflow for problem creation. The medical images used in the benchmark were sourced from publicly available resources online rather than from a dedicated human subjects study. The paper clearly documents the participation of medical experts who contributed to creating the benchmark.

Guidelines:

- The answer NA means that the paper does not involve crowdsourcing nor research with human subjects.
- Including this information in the supplemental material is fine, but if the main contribution of the paper involves human subjects, then as much detail as possible should be included in the main paper.
- According to the NeurIPS Code of Ethics, workers involved in data collection, curation, or other labor should be paid at least the minimum wage in the country of the data collector.

15. **Institutional review board (IRB) approvals or equivalent for research with human subjects**

Question: Does the paper describe potential risks incurred by study participants, whether such risks were disclosed to the subjects, and whether Institutional Review Board (IRB) approvals (or an equivalent approval/review based on the requirements of your country or institution) were obtained?

Answer: [NA]

Justification: The paper does not involve research with human subjects in the traditional sense requiring IRB approval. The medical experts who contributed to dataset creation were collaborators in the research process rather than study participants. Additionally, the paper uses publicly available medical images rather than conducting studies that would expose human participants to risks. No patient data was collected or used in this research, and no interventions or experiments were performed on human subjects that would necessitate IRB approval or risk disclosure.

Guidelines:

- The answer NA means that the paper does not involve crowdsourcing nor research with human subjects.
- Depending on the country in which research is conducted, IRB approval (or equivalent) may be required for any human subjects research. If you obtained IRB approval, you should clearly state this in the paper.

- We recognize that the procedures for this may vary significantly between institutions and locations, and we expect authors to adhere to the NeurIPS Code of Ethics and the guidelines for their institution.
- For initial submissions, do not include any information that would break anonymity (if applicable), such as the institution conducting the review.

16. **Declaration of LLM usage**

Question: Does the paper describe the usage of LLMs if it is an important, original, or non-standard component of the core methods in this research? Note that if the LLM is used only for writing, editing, or formatting purposes and does not impact the core methodology, scientific rigorousness, or originality of the research, declaration is not required.

Answer: [Yes]

Justification: The paper clearly describes the usage of various MLLMs/LLMs as they are central to our research. The methodology sections detail how these models were used, including prompting strategies, evaluation metrics, and performance analysis. Since evaluating these models' in-context learning capabilities is the core purpose of the research rather than just a supplementary element, the paper appropriately documents their usage, specifications, and implementation details.

Guidelines:

- The answer NA means that the core method development in this research does not involve LLMs as any important, original, or non-standard components.
- Please refer to our LLM policy (`https://neurips.cc/Conferences/2025/LLM`) for what should or should not be described.

# Appendix

# Contents

## A   Related Work

In recent years, Large Language Models (LLMs) and Multimodal LLMs (MLLMs) have demonstrated advanced capabilities on medical reasoning tasks. In this section, we provide an overview of key prior works on in-context learning, medical MLLMs, and benchmarking efforts.

**In-Context Learning:** In-context learning (ICL) was popularized by [5] in their paper introducing GPT-3, demonstrating that LLMs can learn to solve tasks at inference time by merely conditioning on a few labeled examples in the input prompt without any gradient-based fine-tuning. This paradigm shift has enabled models to generalize to new tasks at inference time simply from natural language instructions and exemplars alone. The extension of ICL to the vision-language domain was pioneered by Flamingo, a powerful model trained on interleaved sequences of images and text [1]. Flamingo showcased strong few-shot performance on a wide range of visual question answering (VQA) and image captioning tasks by learning entirely from prompts composed of image-text pairs, thereby introducing the first stepping stone towards multimodal ICL.

**MLLMs in Medicine:** Inspired by general-purpose MLLMs like Flamingo [1] and Llava [24], recent works have proposed medical MLLMs capable of handling tasks such as radiology report generation, visual question answering, and medical diagnosis. This includes works like Med-PaLM M [28], Med-Flamingo [25], Llava-Med [20], ChexAgent [7], and BiomedGPT [35]. The ability of these models to perform multimodal in-context learning has not been well studied due to a lack of available benchmarks.

**Benchmarking Multimodal ICL:** Evaluating the ability of MLLMs to effectively learn from multimodal in-context examples at inference time is challenging. In the general domain, several

works have presented approaches for evaluating the ICL capabilities of MLLMs [34, 4, 6]; in particular, [37] recently introduced VL-ICL Bench. The domain of medicine serves as an optimal application domain for multimodal ICL due to the presence of highly-specialized concepts and complex imagery as well as the potential for real-world clinical impact. However, to the best of our knowledge, **no benchmarks have been previously developed to evaluate multimodal ICL capabilities in the medical domain**. Our benchmark SMMILE is designed to bridge this research gap. Prior works in the medical domain evaluated few-shot visual question answering or radiology report generation with benchmarks such as VQA-RAD [14], PathVQA [11], SLAKE [21], or MIMIC-CXR [12]; typically, few-shot examplars in these settings are chosen in an automated fashion via random selection [25, 32]. In contrast, in-context examples included in SMMILE are carefully curated by experts in order to serve as relevant task demonstrations that support the learning of the task at hand.

## B  Dataset Curation

In this section, we provide extended details on our four-step expert-guided data curation procedure. The front-end of our data collection platform consists of a single-page React client that collects each panel's metadata (question, answer, public image URL, specialty, author, order). Once the contributor finishes a problem, the client posts the structured annotations to the back-end. The back-end converts these annotations to an parquet file and uploads the shard to a version-controlled HuggingFace Hub dataset.

### B.1  Step 1: Instructions for Clinical Experts

Clinical experts are provided with detailed instructions covering topic scope, data sourcing, and answer formatting, as shown below.

---

**Instructions for Clinical Experts**

We're excited that you are participating in this research project to create a medical visual question-answering (VQA) benchmark for multimodal AI models! We focus on challenging tasks for which we provide a model with few multimodal context examples, to be followed by a final problem (see below in the visual examples) which on its own is not easy to solve for existing vision-language models (i.e., those of us with access can check with GPT4-V).

Topics: The problems can range across all medical specialties, including radiology images, photographs, pathology slices, ophthalmology imaging etc. Try to focus mostly on 2D images (e.g., slice of CT, Chest X-ray, Photograph etc.), but links to further modalities are welcome (audio, video, sequencing etc.) - as long as they can be referred to via URL.

Data: Do NOT upload any media (e.g., images, videos, audio). Instead, please provide a URL (link) to a publicly available media resource. Do not display any identifiable patient information.

Guidelines: Try to follow a consistent answer format within a given problem - if the problem allows for it. Most importantly, answers must follow a consistent format: "Epidural hematoma, left.", "Subdural hematoma, right." etc. Two in-context examples minimum - 10 maximum.

---

## B.2 Step 2: Homepage Interface

The expert is directed to the homepage interface (Figure 6), where they can initialize a new problem.

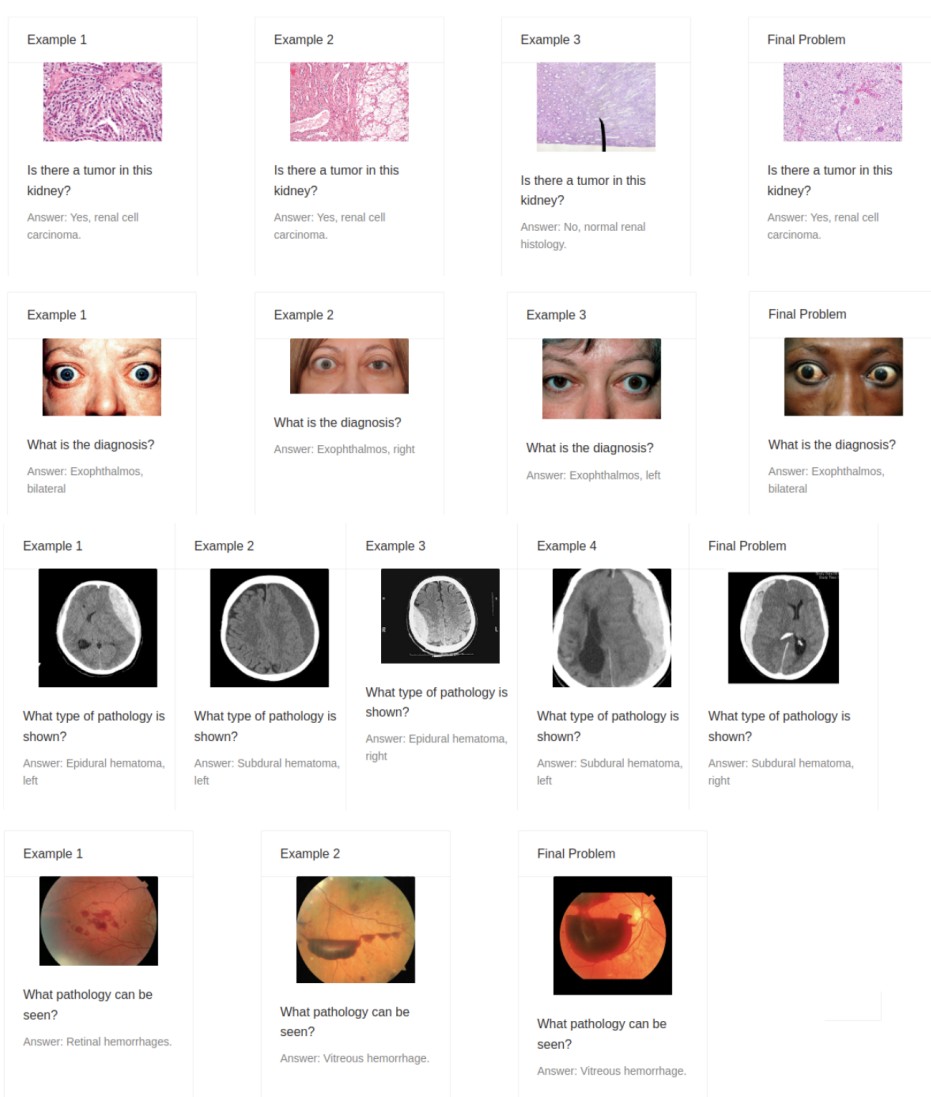

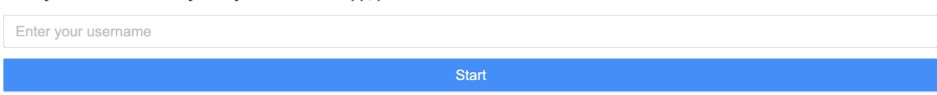

Figure 6: Experts are directed to the homepage interface, visualized here.

### B.3 Step 3: Problem Creation

The problem creation tool is then loaded (Figure 7), where the expert can select the relevant medical specialty as well as add, remove, or reorder panels for in-context examples and the final query.

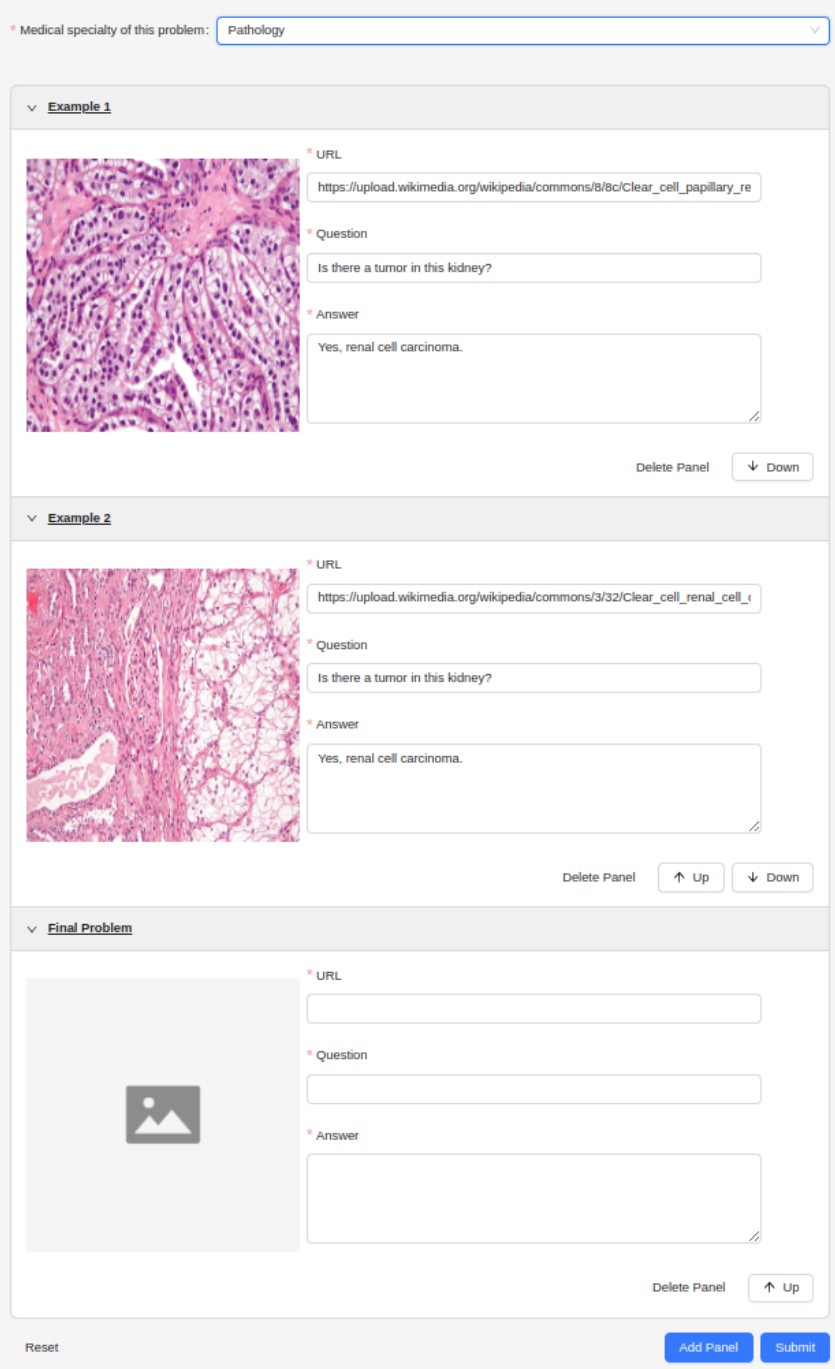

Figure 7: The expert first selects the medical associated with the problem. Then, the expert adds or removes panels corresponding to in-context examples. The expert can also reorder panels to sort the in-context examples and final problem.

## B.4   Step 4: Final Submission

Upon clicking "Submit", the expert is shown an overview of the completed problem for validation or further editing (Figure 8).

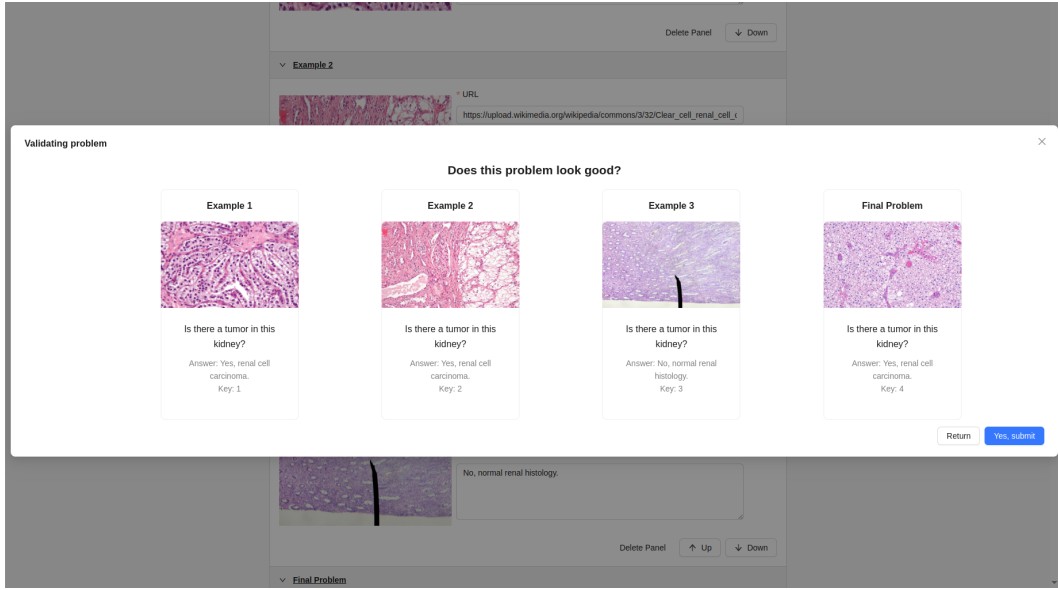

Figure 8: After the expert clicks "Submit", they are presented with an overview of their newly created problem. The expert can then validate the problem or return to the previous screen for more edits.

## C   Descriptive Statistics for SMMILE++

Figure 9 analyzes the composition of SMMILE++ with several descriptive statistics.

## D   Additional Experimental Details

### D.1   Computational Requirements

All experiments with local models were conducted on a research cluster equipped with 8 NVIDIA H200 (141 GB) GPUs. For the larger models (>30B parameters), we used 2-4 GPUs with model parallelism to accommodate memory requirements. The average inference time per sample varied from 3 seconds for the smaller models (0.5B-7B) to 15 seconds for the largest open-source models (70B-90B). Evaluating the entire SMMILE benchmark (111 problems) took between 10 minutes and two hours for a single model configuration, while evaluating the augmented SMMILE++ benchmark (1063 problems) required around 5-10 hours per model. For API-based models (GPT-4o and Claude 3.7 Sonnet), we used their respective APIs with rate limiting considerations, resulting in longer evaluation times.

### D.2   LLM-as-a-Judge Implementation Details

The LLM-as-a-Judge evaluations capture semantic correctness beyond exact string matching, making it particularly valuable for medical reasoning tasks where multiple phrasings might convey the same diagnosis or finding. LLM-as-a-Judge evaluations were performed with Llama3.3 70B, accessed via the Ollama software package[2]. The input prompt is provided below. The output is a binary value, which we multiply by 100 to achieve a final score of either 0 (incorrect) or 100 (correct) for each generated output.

---

[2]Ollama can be accessed at https://github.com/ollama/ollama.

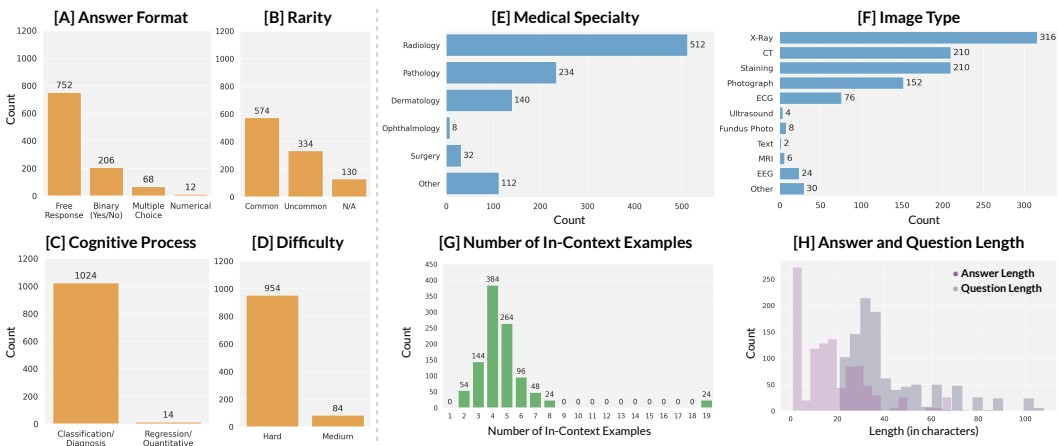

Figure 9: Dataset characteristics. (A–D) Distribution of four key categorical annotations across the unique problems: (A) answer format, (B) rarity of the clinical case based on how often clinicians would experience the medical concepts included in each problem, (C) primary cognitive process required (where reasoning classification is defined by final problem not having direct support in its in-context example set), and (D) rated difficulty. (E–F) Horizontal barplots showing the breakdown of each problem by its main medical specialty (E) and by main image type used (F). (G) Histogram of the number of in-context examples provided per problem. (H) Overlaid histograms of the character-length distributions for questions versus answers. All panels are based on the 1038 problems included in SMMILE++.

---

**LLM-as-a-Judge prompt**

A medical AI model is provided with an image and asked the question "question". The correct answer to this question is: "answer". The AI model outputs "response" as its response. Is the AI model correct? Please output your answer as a single digit, where 1 indicates that the AI model is correct and 0 indicates that the AI model is incorrect with respect to the correct answer. Do not provide anything other than the digit in your response.

---

We opted to run LLM-as-a-Judge evaluations with Llama3.3 70B because (1) Llama3.3 has been shown in prior work [9] to demonstrate strong performance on textual analysis tasks, and (2) Llama3.3 is open-source, generates reproducible results, and does not require payment, ensuring that our benchmark can be useful even in resource-constrained settings. Using a stronger LLM such as GPT-4o results in largely similar results to using Llama3.3, with high interrater agreement observed between the two models (Cohen's kappa = 0.943 across results from six MLLMs in the open-ended setting).

## D.3 Analysis of Prompting Approach

For all MLLMs evaluated in this work, we use a standard input prompt consisting of a system message, in-context examples, and the query image and question. In this section, we provide an analysis of input prompt structure on performance; specifically, we compare our standard prompting approach with Chain-of-Thought (CoT) prompting [29]. CoT prompting operates as follows: for each problem in the SMMILE benchmark, we present a system message and the multimodal in-context examples to the MLLM, followed by a query consisting of an image, a question, and an instruction of the form, "First, explain your reasoning step-by-step by referring to the provided image. Then, answer the question." Results on the SMMILE benchmark (open-ended ICL setting) are summarized in Table 4.

Across the evaluated models, we see that performance improvements afforded by CoT are minor, and in fact, multiple models exhibit degraded performance when using CoT prompting. In particular, we observe substantial drops in performance for LLaVA-OneVision-0.5B and LLaVA-v1.5-13B. Further analysis demonstrates that both models exhibit high rates of malformed outputs (e.g. outputs such

Table 4: Here, we analyze the effects of prompt structure on SMMILE benchmark performance (open-ended ICL setting) across a sample of 5 MLLMs. We consider two options for prompt structure: standard prompting and chain-of-thought (CoT) prompting.

| Model | Standard Prompting | CoT Prompting |
|---|---|---|
| LLaVA-OneVision-0.5B | **21.63** $\pm_{4.00}$ | 7.23 $\pm_{2.53}$ |
| Qwen2.5-VL-3B | **33.58** $\pm_{4.09}$ | 27.63 $\pm_{4.91}$ |
| LLaVA-Med-7B | 10.19 $\pm_{3.06}$ | **12.45** $\pm_{3.14}$ |
| Qwen2.5-VL-7B | 29.58 $\pm_{4.63}$ | **29.95** $\pm_{4.75}$ |
| LLaVA-v1.5-13B | **20.91** $\pm_{3.49}$ | 15.75 $\pm_{3.44}$ |

as "ooooooooooo..." or "( and ( ( (and (...") , suggesting that these models are unable to effectively respond to the prompt. Additionally, using CoT prompting results in substantial increases in inference time, particularly for the Qwen model family. As a result, we utilize the standard prompting approach for all evaluations in this work.

## E   Extended Fine-Grained Analysis for SMMILE

Figure 10 provides an extended fine-grained analysis of MLLM performance on the SMMILE benchmark. Figure 11 analyzes MLLM performance on the SMMILE benchmark stratified by number of in-context examples provided to the model.

## F   Extended Fine-Grained Analysis for SMMILE++

Figure 12 provides a fine-grained analysis of MLLM performance on the SMMILE++ benchmark. Figure 13 analyzes MLLM performance on the SMMILE++ benchmark stratified by number of in-context examples provided to the model.

## G   Limitations and Future Work

We note several key directions for future work:

1. *Scale.* Crowdsourcing or synthetic augmentation could help expand coverage across the thirteen data modalities included in SMMILE.

2. *Modalities.* Time-series signals, volumetric scans, genomics, and structured EHR fields are not currently included.

3. *Expert diversity.* Current contributors may not capture all specialties or practice settings; future work can look at recruiting a larger diversity of expert contributors.

4. *Task scope.* The benchmark centers on diagnosis; extensions to treatment planning, prognosis, and longitudinal reasoning would help cover other aspects of clinical workflows.

Nonetheless, SMMILE exposes concrete weaknesses in today's MLLMs when applied to medical scenarios and supplies the community with a rigorous, extensible framework for further research.

## H   Licensing Considerations

This benchmark includes question-answer pairs generated by medical experts, licensed under CC BY 4.0. SMMILE is available at https://smmile-benchmark.github.io.

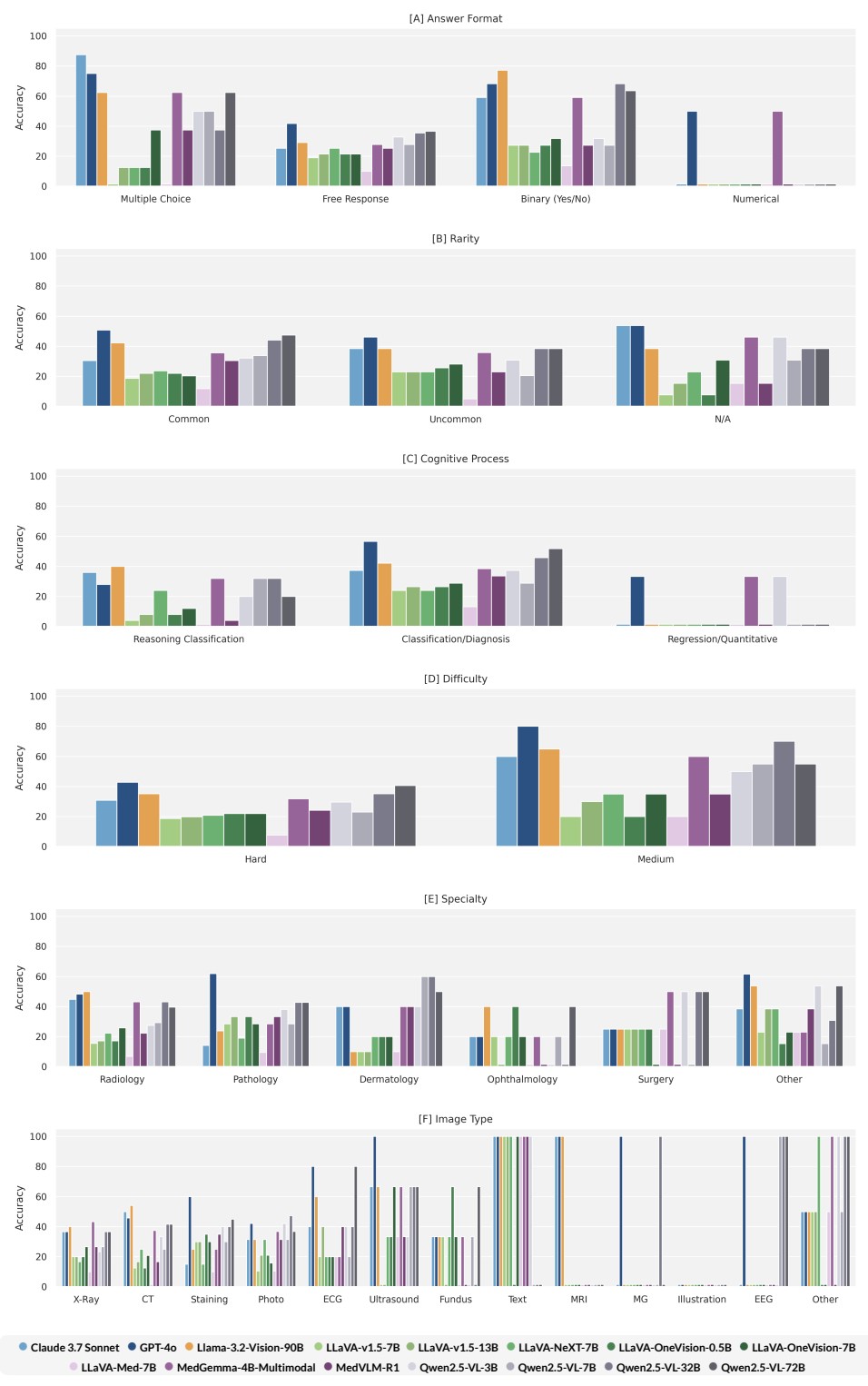

Figure 10: We provide a fine-grained breakdown of MLLM performance on the SMMILE benchmark. We report performance stratified by answer format (Panel A), rarity (Panel B), cognitive process (Panel C), difficulty (Panel D), medical specialty (Panel E), and image type (Panel F). Here, we focus on open-ended evaluations, and the y-axis represents prediction accuracy as computed by the LLM-as-a-Judge approach. The acronym MG refers to Mammograms.

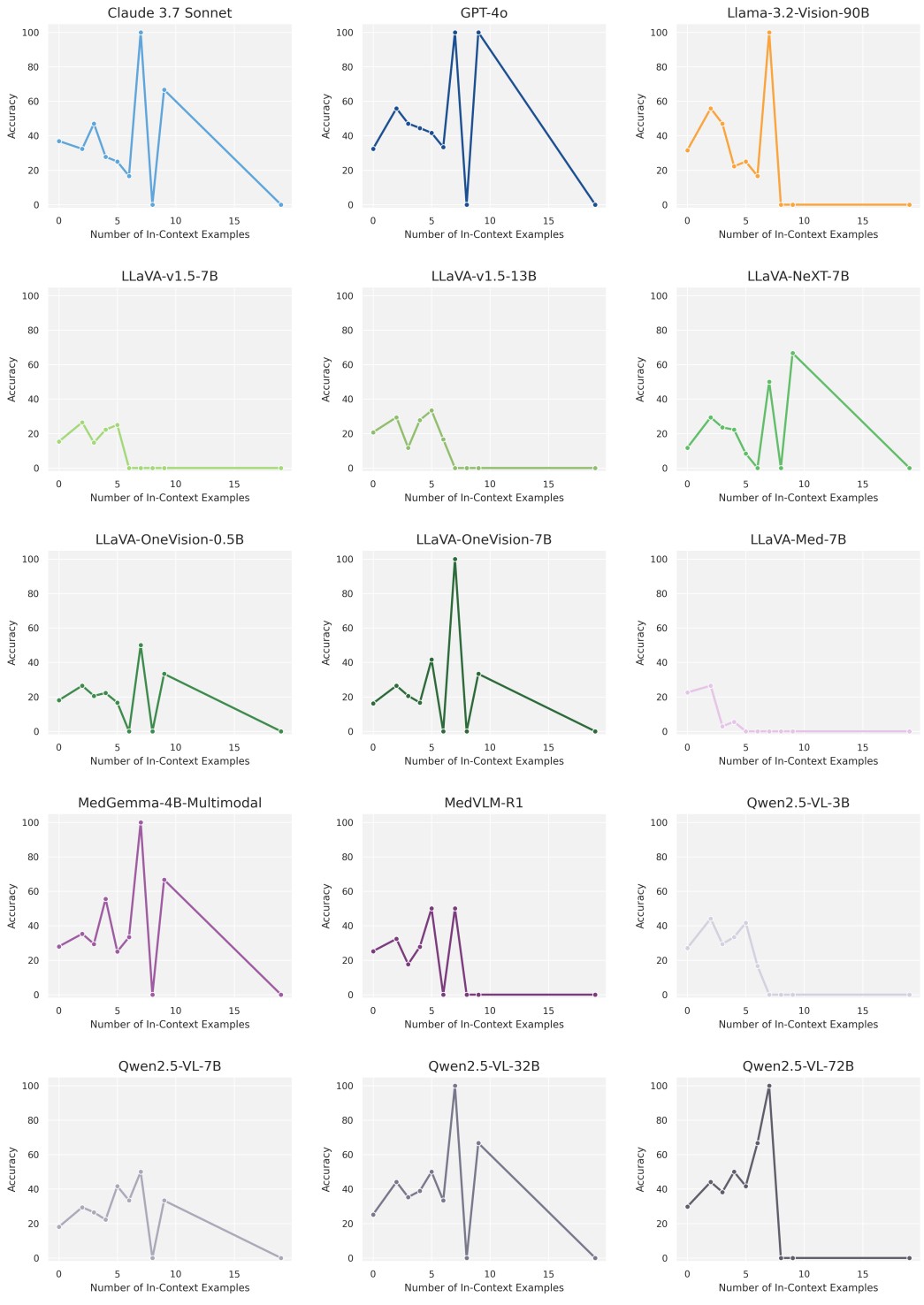

Figure 11: We analyze MLLM performance on the SMMILE benchmark stratified by number of in-context examples provided to the model. Here, we focus on open-ended evaluations, and the y-axis represents prediction accuracy as computed by the LLM-as-a-Judge approach.

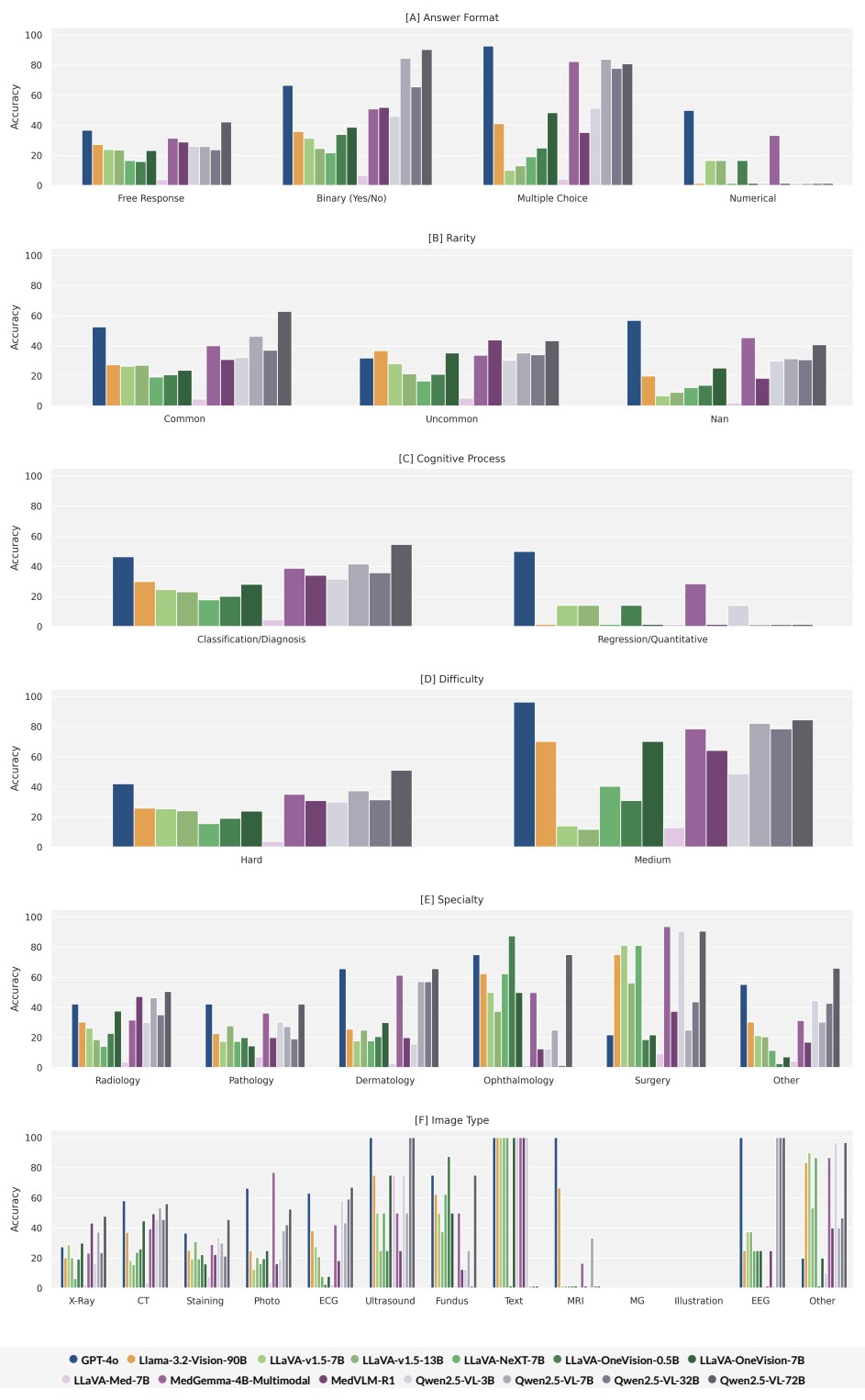

Figure 12: We provide a fine-grained breakdown of MLLM performance on the SMMILE++ benchmark. We report performance stratified by answer format (Panel A), rarity (Panel B), cognitive process (Panel C), difficulty (Panel D), medical specialty (Panel E), and image type (Panel F). Here, we focus on open-ended evaluations, and the y-axis represents prediction accuracy as computed by the LLM-as-a-Judge approach. The acronym MG refers to Mammograms.

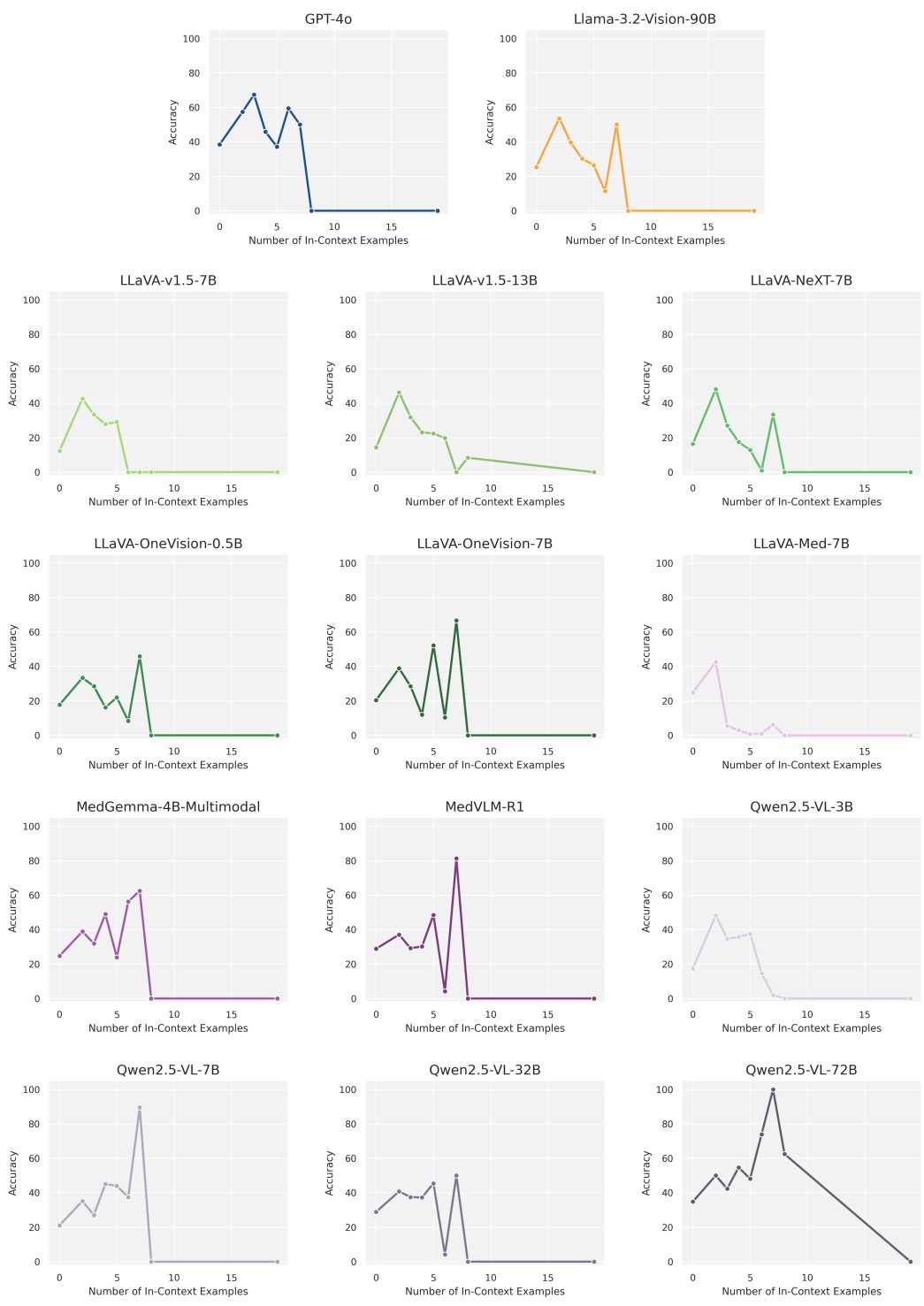

Figure 13: We analyze MLLM performance on the SMMILE++ benchmark stratified by number of in-context examples provided to the model. Here, we focus on open-ended evaluations, and the y-axis represents prediction accuracy as computed by the LLM-as-a-Judge approach.

