# OpenReview forum: "SMMILE: An expert-driven benchmark for multimodal medical in-context learning"
_NeurIPS.cc/2025/Datasets_and_Benchmarks_Track — NeurIPS 2025 Datasets and Benchmarks Track poster_

### Official Review · Reviewer_avhB · 2025-07-02

**Rating:** 5
**Confidence:** 4

**Summary:**

The paper introduces SMMILE, **the first expert-curated benchmark for evaluating multimodal in-context learning (ICL) in the medical domain**. This dataset consists of 517 VQA samples that are structured into 111 in-context learning problems, with an augmented variant SMMILE++ containing 1,063 permuted problems. This dataset is created by 11 clinical experts across 6 specialties and 13 types of image, each problem includes a multimodal query and two or more in-context examples that serve as task demonstrations. The authors evaluate 15 diverse VLMs, including 13 open-weight models and 2 closed, API-based frontier models, and find that most models exhibit limited ICL capabilities. The study points out key challenges that include recency bias, poor handling of numerical answers, and modality-specific weaknesses, offering a rigorous, expert-driven framework for future research of medical MLLMs.

**Dataset Code Accessibility:**

Yes

**Dataset Code Comments:**

The authors provided information about the dataset and reproduced experiments on their anonymous GitHub.

**Ethical Considerations:**

No, there are no or only very minor ethics concerns

**Final Justification:**

I decided to keep my current positive scores to authors.

**Limitations Weaknesses:**

Reviewer has the following suggestions (not weaknesses) to make the benchmark more informative.

- **Integrating Retrieval-Augmented Generation (RAG) as a part of input context**: Rather than just providing a few input images and corresponding outputs, it would be interesting to see if using information from medical RAG on the topic question can enhance model performance, especially with medical MLLM. These retrieval ones can be in the text form extracted from guidelines, etc.
- **Prompting for Chain-of-Thought Reasoning**: Instead of prompting models to output an answer directly, prompt them to:
***"First explain your reasoning, then provide the answer."***. For instance,

```
Image: [Chest X-ray]
Question: What is the most likely diagnosis?
Instruction: First, explain your reasoning step-by-step by referring to the provided image. Then, state your final diagnosis.
```
Such a prompt can encourage the models to: (i) verbalize intermediate reasoning steps (e.g., image observations, differential diagnoses); (ii) reduce hallucinations by making logic explicit, and (iii) improve interpretability for human reviewers. It would be great if authors could provide experimental results with the above prompt on a few models (like LLaVa-Med) and provide insights.

**Strengths Contributions:**

Reviewer found the following strength points of the papers:

- **First Expert-Curated Benchmark for Multimodal ICL in Medicine**: According my knowledge, this is the fist benchmark created by experts specifically designed to evaluate multimodal in-context learning (ICL) in the medical domain, filling a critical gap in both the ICL and medical AI research. Unlike prior few-shot evaluations that randomly retrieved examples from the same benchmark, this dataset emphasises task-specific, multimodal ICL.

- **Comprehensive Evaluation of Models:** The authors conduct a systematic evaluation of 15 SOTA VLMs, revealing that most models struggle with multimodal ICL, and that domain-specific models do not consistently outperform general models. These findings highlight limitations in current MLLM capabilities. Analyze also covers different aspects such as injecting noise into input (Section 4.1), analyzing Example Order (Section 4.2), etc.

In summary, this benchmark is crucial for evaluating the capabilities of current medical multi-modal LLMs in real-world scenarios.

---

> ### Author Rebuttal · Authors · 2025-07-31
>
> We thank Reviewer AVHB for reviewing our work and providing helpful feedback.
>
> > **[Q1] RAG as part of input context. “Rather than just providing a few input images and corresponding outputs, it would be interesting to see if using information from medical RAG on the topic question can enhance model performance, especially with medical MLLM. These retrieval ones can be in the text form extracted from guidelines, etc.”**
>
> Thank you for this suggestion - we agree that exploring the role of RAG techniques on the performance of MLLMs in the medical domain is an interesting research direction. However, the key goal of our benchmarking study is to evaluate the extent to which MLLMs can learn to perform challenging medical tasks when provided with relevant task demonstrations in the form of multimodal ICL examples. In order to accurately characterize ICL capabilities in isolation, the MLLMs in our benchmarking study were not provided with access to external resources and were expected to learn the task solely from the expert-constructed multimodal ICL examples. As a result, RAG is out of scope for our current benchmarking study, as the incorporation of external knowledge would make it challenging to effectively understand and analyze MLLM ICL capabilities. However, we note that to some extent, our task setup can be viewed as a special case of RAG, where we assume that we have perfect retrieval of a few highly-relevant examples; in Section 4.1, we show that the presence of irrelevant examples (which can be viewed as imperfect example retrieval) often leads to degraded performance.
>
> We hope that future works will build on the SMMILE benchmark by exploring techniques like RAG. We note that text-based RAG (e.g. with medical guidelines) is unlikely to be useful in this setting as the majority of the relevant information needed to answer questions lies within images and not the question itself; for example, questions are often of the form “What is shown in the image?” or “What type of pathology is shown?” (see Figure 1). However, exploring multimodal RAG techniques would certainly be an interesting direction for future work.
>
> > **[Q2] Chain-of-Thought Reasoning. “Instead of prompting models to output an answer directly, prompt them to: "First explain your reasoning, then provide the answer.” Such a prompt can encourage the models to: (i) verbalize intermediate reasoning steps (e.g., image observations, differential diagnoses); (ii) reduce hallucinations by making logic explicit, and (iii) improve interpretability for human reviewers. It would be great if authors could provide experimental results with the above prompt on a few models (like LLaVa-Med) and provide insights.”**
>
> Below, we evaluate the effects of Chain-of-Thought prompting on the multimodal ICL capabilities of five MLLMs. For each problem in the SMMILE benchmark, we present the multimodal in-context examples to the MLLM, followed by a query consisting of an image, a question, and an instruction of the form, "First, explain your reasoning step-by-step by referring to the provided image. Then, answer the question."
>
> Results on the SMMILE benchmark (open-ended ICL setting) are summarized below:
>
> |                      | Default Prompting | CoT Prompting |
> |----------------------|-------------------|---------------|
> | LLaVA-Onevision-0.5B | 21.63 ± 4.00      | 7.23 ± 2.53   |
> | Qwen2.5-VL-3B        | 33.58 ± 4.09      | 27.63 ± 4.91  |
> | Qwen2.5-VL-7B        | 29.58 ± 4.63      | 29.95 ± 4.75  |
> | LLaVA-Med            | 10.19 ± 3.06      | 12.45 ± 3.14  |
> | LLaVA-v1.5-13B       | 20.91 ± 3.49      | 15.75 ± 3.44  |
>
> Key takeaways from this evaluation are summarized below:
> - Across the evaluated models, we see that performance improvements afforded by CoT are minor, and in fact, multiple models exhibit degraded performance when using CoT prompting. In particular, we observe substantial drops in performance for LLaVA-Onevision-0.5B and LLaVA-v1.5-13B. Further analysis demonstrates that both models exhibit high rates of malformed outputs (e.g. outputs such as "ooooooooooo..." or "( and ( ( (and (..."), suggesting that these models are unable to effectively respond to the prompt.
> - Across the evaluated models, we see that incorporating CoT prompting is not sufficient for fully solving the SMMILE benchmark.
> - Using CoT prompting results in substantial increases in inference time, particularly for the Qwen models. We observe inference time increases on the order of 2-3x per sample.
>
> We agree with the reviewer that studying the effects of prompt variations on model performance is an important research direction, and the high-quality nature of benchmark enables such analyses to be performed and meaningful conclusions to be drawn. We will provide support for prompt variation experiments in our benchmark and analysis toolkit, and we will be sure to include an extended analysis of prompt variations in the final version of our paper.
>
> We again thank Reviewer AVHB for their review of our manuscript and their positive overall assessment of our work. We hope that the above responses adequately address all concerns.

---

> > ### Comment · Reviewer_avhB · 2025-08-05
> > **Response to Authors During Discussion**
> >
> > Thanks authors for your responses. I decided to keep my positive rating scores!

---

### Official Review · Reviewer_4McS · 2025-07-03

**Rating:** 5
**Confidence:** 4

**Summary:**

This paper introduces SMMILE, a benchmark designed to evaluate multimodal in-context learning (ICL) capabilities of VLMs in medical settings. The benchmark was collaboratively created by 11 clinical experts and consists of 517 visual question-answering samples structured into 111 ICL problems across various medical specialties. The authors evaluated 14 VLMs and found that most models show limited improvement from multimodal ICL in medicine, with only two achieving meaningful gains. The study also reveals significant biases (recency bias and noisy examples).

**Dataset Code Accessibility:**

Yes

**Dataset Code Comments:**

Dataset is publicly available.

**Ethical Considerations:**

No, there are no or only very minor ethics concerns

**Final Justification:**

Thanks for the clarification. My questions are resolved. I'll keep the rating.

**Limitations Weaknesses:**

1. The benchmark contains only 111 problems, which is relatively small for a comprehensive benchmark.
2. Exact match evaluation showing 0% for many models suggests potentially too strict criteria. Heavy reliance on LLM-as-judge evaluation may introduce its own biases.
3. Could the authors provide human expert performance as a reference standard to better contextualize model performance?

**Strengths Contributions:**

1. This benchmark is dedicated to multimodal medical ICL, filling an important gap in the health AI domain.
2. The benchmark design is expert-driven with comprehensive quality control, involving 11 international clinical experts who created medically authentic problems rather than randomly sampled ones.
3. The authros evaluate 14 diverse VLMs including both open-source and closed-source models across different medical specialties. Statistical tests with bootstrap were performed.
4. The data curation and model evaluation are rigorous with detailed procedures described in the manuscript. The paper provides insightful analysis of biases, failure modes, and performance across different medical contexts.

---

> ### Author Rebuttal · Authors · 2025-07-31
>
> We thank Reviewer 4MCS for reviewing our work and providing helpful feedback.
>
> > **[Q1] Size of Benchmark. “The benchmark contains only 111 problems, which is relatively small for a comprehensive benchmark.”**
>
> Thank you for raising this point. We emphasize that SMMILE is indeed a comprehensive benchmark, including 517 question-image-answer triplets from 6 medical specialties and 13 imaging modalities. Naturally, expert-driven benchmarks are by design challenging to scale, yet in contrast to automatically-curated medical datasets that are commonplace, SMMILE offers the critical advantage of high sample quality. We also note that the size of the SMMILE benchmark offers two key advantages. First, manual quality control is made possible; two authors verified every problem in the SMMILE benchmark, yielding a high-quality dataset that enables accurate and meaningful conclusions to be drawn from model benchmarking. Second, compute-intensive MLLMs can be benchmarked rapidly, making MLLM evaluation feasible even in resource-constrained settings (Appendix C.1 provides further discussion on compute requirements). In contrast, large-scale benchmarks are often prohibitively expensive, particularly when evaluating API-based models, often causing users to instead evaluate on smaller subsets.
>
> In addition to SMMILE, we also introduce SMMILE++ in this work, a substantially larger benchmark with over 1000 ICL problems created by permuting in-context examples from a subset of problems in SMMILE.
>
> > **[Q2] Metrics. “Exact match evaluation showing 0% for many models suggests potentially too strict criteria. Heavy reliance on LLM-as-judge evaluation may introduce its own biases.”,“Could the authors provide human expert performance as a reference standard to better contextualize model performance?”**
>
> We thank the reviewer for raising these points about exact match rigidity and the need for clinician evaluations. Our evaluation deliberately employs multiple complementary metrics. While exact match (EM)  appears rigid (showing 0% for several models), this reveals a critical limitation of current MLLMs in generating precisely formatted medical responses. Higher EM scores in the in-context learning setting show that models can learn to mimic the concise formatting of in-context examples, yet still fall short of providing correct answers. We complement this with LLM-as-a-Judge for evaluating semantic correctness.
>
> Following your suggestion, we added human expert evaluation to complement our metrics. Due to resource constraints, we focused the human evaluation on the SMMILE dataset, recruiting 5 clinical experts who independently provided binary ratings of model responses. Each response was evaluated by 2 different clinicians in both in-context learning (ICL) and 0-shot settings. We report the expert ratings in the table below. We note that perfect inter-rater agreement (100%, κ = 1.0) was found in the ICL setting on all models, and inter-rater agreement ranging from 98.2% to 100% (κ = 0.91 to κ = 1.0) in the 0-shot setting.
>
> | Model                    | Average Expert Rating, 0-shot (%) | Expert Rating, ICL (%)* |
> |--------------------------|-----------------------------------|------------------------|
> | Claude 3.7 Sonnet        | 39.19                             | 44.14                  |
> | GPT-4o                   | 33.33                             | 43.24                  |
> | Llama-3.2-Vision-90B     | 36.94                             | 33.33                  |
> | LLaVA-v1.5-7B           | 17.12                             | 21.62                  |
> | LLaVA-v1.5-13B          | 22.52                             | 26.13                  |
> | LLaVA-NeXT-7B           | 13.51                             | 33.33                  |
> | LLaVA-OneVision-0.5B    | 19.82                             | 18.02                  |
> | LLaVA-OneVision-7B      | 25.23                             | 27.03                  |
> | LLaVA-Med-7B            | 22.52                             | 10.81                  |
> | MedGemma-4B-Multim.     | 27.03                             | 32.43                  |
> | MedVLM-R1               | 25.23                             | 24.32                  |
> | Qwen2.5-VL-3B           | 23.42                             | 23.42                  |
> | Qwen2.5-VL-7B           | 11.71                             | 27.03                  |
> | Qwen2.5-VL-32B          | 21.62                             | 32.43                  |
> | Qwen2.5-VL-72B          | 26.13                             | 31.52                  |
>
> We again thank Reviewer 4MCS for their review of our manuscript and their positive overall assessment of our work. We hope that the above responses adequately address all concerns.

---

### Official Review · Reviewer_7LMU · 2025-07-04

**Ethics Flags:** Data privacy, copyright, and consent
**Rating:** 4
**Confidence:** 4

**Summary:**

In this work, the authors study and investigate the multimodal medical in-context learning problem. They also collected a Stanford Multimodal Medical In-context Learning benchmark, where 11 clinical experts created the ICL problems. Then, 14 diverse vision language models are evaluated on this new benchmark. The results show only GPT-4o and Qwen 2.5-VL-32B achieved double-digit improvements over their zero-shot accuracy.

**Additional Feedback:**

NA

**Dataset Code Accessibility:**

Yes

**Dataset Code Comments:**

NA

**Ethical Considerations:**

No, there are no or only very minor ethics concerns

**Final Justification:**

Thanks for the authors' feedback. This reviewer had carefully read the responses and other reviews. They have addressed my concerns. Thus, this reviewer keeps the rating.

**Limitations Weaknesses:**

1. While the benchmark is curated by clinical experts, some of the in-context learning problems appear to be solvable by small models without truly leveraging contextual learning, such as disease types or image modalities. Some of questions may be simplistic or not closely aligned with the decision-making processes in real clinical settings, potentially limiting the benchmark’s discriminative contribution.

2. The study would benefit from deeper ablation or investigation into key variables influencing in-context learning performance, such as the number of support examples, modality combinations, or prompt structure. A more systematic analysis could offer valuable insights into how different configurations impact model behavior and help inform better design of ICL tasks in the medical domain.

**Strengths Contributions:**

1. The paper studies an important and underexplored multimodal in-context learning in medicine, which is highly relevant for real-world clinical decision-making. The involvement of a team of clinical experts in curating the benchmark ensures the tasks are grounded in authentic medical scenarios.
2. The paper is well-structured, with a clear and logical flow covering dataset construction, model evaluation, and analysis of key findings.
3. Figures and tables are well-designed and contribute significantly to the clarity and interpretability of the experimental results.

---

> ### Author Rebuttal · Authors · 2025-07-31
>
> We thank Reviewer 7LMU for reviewing our work and providing helpful feedback.
>
> > **[Q1] Question Difficulty. “While the benchmark is curated by clinical experts, some of the in-context learning problems appear to be solvable by small models without truly leveraging contextual learning, such as disease types or image modalities. Some of the questions may be simplistic or not closely aligned with the decision-making processes in real clinical settings, potentially limiting the benchmark’s discriminative contribution.”**
>
> Thank you for bringing this up. This observation actually highlights one of our key findings: while individual problems may appear straightforward to clinical experts or even non-experts (such as counting neutrophils on a microscopy image), our results demonstrate that current MLLMs struggle significantly: the best-performing model without context examples (Claude 3.7 Sonnet) achieves only 37.2% accuracy, and with in-context learning, 7 out of 15 models fail to exceed random baseline performance (27.86%). Furthermore, our benchmark problems were curated by practicing clinicians and clinician-scientists to reflect authentic diagnostic scenarios they encounter, where physicians must integrate visual findings with contextual knowledge from similar cases as well as differential diagnoses—a core component of clinical decision-making that directly parallels the in-context learning paradigm we evaluate.
>
> >**[Q2] Additional Analysis. “The study would benefit from deeper ablation or investigation into key variables influencing in-context learning performance, such as the number of support examples, modality combinations, or prompt structure. A more systematic analysis could offer valuable insights into how different configurations impact model behavior and help inform better design of ICL tasks in the medical domain.”**
>
> Thank you for this suggestion. Our submitted manuscript included several systematic ablations and fine-grained analyses that explored the effects of various design choices on multimodal ICL performance. In response to your suggestion, we have extended these analyses as well as incorporated several additional ablations, which we will incorporate into the final version of our paper. Below, we provide key takeaways from these experiments.
>
> **[Evaluation 1 - Analyzing Example Quality]** Section 4.1 of our submitted paper explores the effects of noisy or irrelevant in-context examples on MLLM performance. We create two perturbed versions of the SMMILE dataset as follows: (1) SMMILE-Random-Noise: For each sample in SMMILE, we add a random image-question-answer triplet from the dataset to the in-context example set. (2) SMMILE-Targeted-Noise: For each sample in SMMILE, we add an image-question-answer triplet from the dataset that shares the same specialty as the sample. Below, we extend the analysis provided in Section 4.1 by evaluating a larger suite of models (9 MLLMs) across these perturbed variants of SMMILE.
>
> |                      | SMMILE          | SMMILE-Random-Noise | SMMILE-Targeted-Noise |
> |----------------------|-----------------|---------------------|-----------------------|
> | LLaVA-OneVision-0.5B | **21.63 ±4.00** | 19.41 ±3.50         | 21.35 ±3.83           |
> | Qwen2.5-VL-3B        | **33.58 ±4.09** | 30.40 ±4.65         | 30.37 ±4.73           |
> | LLaVA-v1.5-7B        | **18.72 ±3.40** | 17.95 ±3.87         | 14.80 ±3.31           |
> | LLaVA-OneVision-7B   | **24.25 ±3.81** | 21.90 ±3.86         | 23.04 ±3.82           |
> | LLaVA-NeXT-7B        | 23.66 ±3.90     | 17.77 ±3.26         | **24.38 ±3.99**       |
> | LLaVA-Med-7B         | **10.19 ±3.06** | 4.88 ±2.07          | 1.88 ±1.32            |
> | Qwen2.5-VL-7B        | 29.58 ±4.63     | **33.11 ±3.92**     | 31.92 ±4.01           |
> | LLaVA-v1.5-13B       | **20.91 ±3.49** | 18.87 ±3.80         | 16.14 ±3.40           |
> | Qwen2.5-VL-32B       | **41.79 ±4.73** | 39.60 ±4.60         | 39.10 ±4.48           |
> | Average              | 24.92           | 22.65               | 22.55                 |
>
> We observe that the inclusion of just one noisy sample in the in-context example list can impair performance, with most models exhibiting performance degradations on both SMMILE-Random-Noise (9.1% relative decrease from SMMILE on average) and SMMILE-Targeted-Noise (9.5% relative decrease from SMMILE on average). Targeted noise contributes to slightly lower performance than random noise on average, suggesting that even targeted, specialty-based selection of in-context examples can impair performance if the selected examples are not effective demonstrations of the task at hand. Importantly, the effects of noise are model-specific; the presence of noisy in-context examples affects each model in differing ways, leading to substantial changes in the final rankings. Our results demonstrate the critical need for high-quality, manually-curated benchmarks for evaluating in-context abilities of MLLMs in the medical setting, as the presence of noisy or irrelevant samples in the in-context example set can prevent developers from accurately understanding the capabilities of trained models.
>
> **[Evaluation 2 - Analyzing Example Order]** Section 4.2 of our submitted paper evaluates the effects of example order on MLLM performance. Specifically, we investigate the extent to which (a) the first in-context example and (b) the last in-context example influence MLLM predictions. We extend the analysis provided in Section 4.2 by perturbing example orderings. Specifically, we first filter the SMMILE dataset to a subset of 69 problems where at least one in-context example has an identical answer to the query question; then, we modify the ordering of the in-context example list such that the placement of examples with identical answers can be explicitly controlled. Given this setup, we evaluate performance of the same 9 MLLMs as the previous evaluation.
>
> We observe substantial performance degradations (absolute decrease of up to 47%) when the answer to the first in-context example matches the answer to the query question. On the other hand, we observe substantial performance improvements (absolute improvement of up to 71%) when the answer to the last in-context example matches the answer to the query question. These trends hold for all nine MLLMs evaluated in this setting, which consist of varied architectures and parameter counts. Importantly, our finding suggests that MLLMs are affected by recency bias, where placing the most relevant in-context examples (i.e. those that share answers with query question) later in the list can lead to improved performance.
>
> **[Evaluation 3 - Number of ICL Examples]** In our submitted paper, we provided analysis on the number of ICL examples in Section 3.3, which we reproduce here. Figure 4 (Panel C) evaluates the effect of the number of ICL examples on MLLM performance. For all evaluated models, providing two ICL examples leads to substantial improvements in performance over the zero-shot setting. However, trends become more variable as the number of ICL examples increases. In particular, we observe that increasing the number of ICL examples is not consistently correlated with stronger performance; in particular, all models exhibit substantial performance degradations that often dip below zero-shot performance. These results suggest that existing MLLMs may be unable to perform ICL tasks when provided with lengthy inputs consisting of multiple interleaved image-text pairs.
>
> **[Evaluation 4 - Prompt Structure]** In response to your suggestion on prompt structure, we evaluate the effects of Chain-of-Thought prompting on the multimodal ICL capabilities of five MLLMs. For each problem in the SMMILE benchmark, we present the multimodal in-context examples to the MLLM, followed by a query consisting of an image, a question, and an instruction of the form, "First, explain your reasoning step-by-step by referring to the provided image. Then, answer the question." Results on the SMMILE benchmark (open-ended ICL setting) are summarized below:
>
> |                      | Default Prompting | CoT Prompting |
> |----------------------|-------------------|---------------|
> | LLaVA-Onevision-0.5B | 21.63 ± 4.00      | 7.23 ± 2.53   |
> | Qwen2.5-VL-3B        | 33.58 ± 4.09      | 27.63 ± 4.91  |
> | Qwen2.5-VL-7B        | 29.58 ± 4.63      | 29.95 ± 4.75  |
> | LLaVA-Med            | 10.19 ± 3.06      | 12.45 ± 3.14  |
> | LLaVA-v1.5-13B       | 20.91 ± 3.49      | 15.75 ± 3.44  |
>
> Across the evaluated models, we see that performance improvements afforded by CoT are minor, and in fact, multiple models exhibit degraded performance when using CoT prompting. In particular, we observe substantial drops in performance for LLaVA-Onevision-0.5B and LLaVA-v1.5-13B. Further analysis demonstrates that both models exhibit high rates of malformed outputs (e.g. outputs such as "ooooooooooo..." or "( and ( ( (and (..."), suggesting that these models are unable to effectively respond to the prompt.
>
> We again thank Reviewer 7LMU for their review of our manuscript and their positive overall assessment of our work. We hope that the above responses adequately address all concerns.

---

> > ### Comment · Reviewer_7LMU · 2025-08-07
> > **Responses**
> >
> > Thanks for the authors' feedback. This reviewer had carefully read the responses and other reviews. They have addressed my concerns. Thus, this reviewer keeps the rating.

---

### Official Review · Reviewer_gYPq · 2025-07-06

**Ethics Flags:** Data privacy, copyright, and consent
**Rating:** 5
**Confidence:** 5

**Summary:**

- The paper presents SMMILE (Stanford Multimodal Medical In-context Learning benchmark), which is the first dedicated benchmark to evaluate multimodal ICL in medicine.
- It introduces two benchmark tasks as part of a new multimodal in-context learning (ICL) benchmark designed specifically for the medical domain (open-ended and close-ended questions and answers).
- A team of 11 clinical experts curated a set of multimodal tasks composed of image–text pairs with varying levels of complexity and relevance, including 2 medical students.
- The study evaluates 14 state-of-the-art VLMs on their ability to perform ICL in the medical domain, rather than relying on zero-shot or fine-tuned learning settings.
- The results show that current VLMs demonstrate limited gains from multimodal ICL
- The benchmark exposes important limitations and biases in the ability of current multimodal AI models to adapt to new medical tasks via in-context learning.

**Dataset Code Accessibility:**

Partly

**Dataset Code Comments:**

I checked the hugging face website and the dataset is available there. However I was unable to find the toolkit for model assessment.

**Ethical Comments:**

This is primarily related to the fact that the authors rely on the use of URLs to include the images - which may be subject to additional consent or copyright issues.

**Ethical Considerations:**

Yes, there are ethics concerns that require attention by the authors

**Final Justification:**

The authors addressed all of my questions and comments adequately. Hence I would like to raise my score.

**Limitations Weaknesses:**

- The paper does not clearly define what is meant by tasks being “difficult for state-of-the-art LLMs to answer,” and it is unclear what criteria were used to determine difficulty during dataset review.
- The open-ended task is evaluated using exact match or by relying on an LLM-as-a-judge. However, this may not adequately capture semantic similarity or partial correctness, and alternative metrics like BERTScore could provide more nuanced assessments.
- The use of exact match as the primary evaluation metric may be too rigid and not well suited for evaluating generative responses in complex medical tasks, as evident by the 0.0% performance of several models.
- Have the authors considered asking the clinicians to evaluate model responses?
- The phrasing “to quantify uncertainty in our metrics” is not precise.
- The evaluation framework only employs LLaMA as the LLM judge, without justification for excluding other strong models like GPT-4 that might offer more robust judgments.
- There is ambiguity regarding the respective roles of medical students in the annotation and validation process, which could affect the reliability of the data - can they be considered as experts? Also inter-rater reliability assessment may be necessary for the subjective assessments.
- The paper lacks details about the tools and technologies used to build the data collection platform, as well as information on the time it takes clinicians to construct the samples

**Strengths Contributions:**

- The contributions are substantial and include: (i) the first systematic evaluation of multimodal in-context learning in the medical domain, (ii) the release of a curated dataset of expert-annotated multimodal examples with graded task difficulty, and (iii) a complete evaluation framework with baseline models and an open-source analysis toolkit.
- The writing is clear, well-structured, and accessible, making the methodology and results easy to follow.
- The paper provides a very detailed and transparent analysis supported by comprehensive statistics and empirical findings, especially for dataset construction.

---

> ### Author Rebuttal · Authors · 2025-07-31
>
> We thank Reviewer GYPQ for reviewing our work and providing helpful feedback.
>
> > **[Q1]  Difficulty of SMMILE questions. “The paper does not clearly define what is meant by tasks being “difficult for state-of-the-art LLMs to answer,” and it is unclear what criteria were used to determine difficulty during dataset review.”**
>
> During dataset curation, clinical experts evaluated problem difficulty with respect to current MLLM capabilities by considering: (1) whether the problem required complex multimodal reasoning beyond simple visual pattern matching, (2) the degree of specialized medical knowledge needed, and (3) to what extent successful answers would likely require leveraging the provided in-context examples rather than relying solely on pre-trained knowledge.
>
> Our difficulty annotations are empirically validated by the actual performance results shown in Figure 3D, where problems labeled as "Hard" consistently demonstrated lower accuracy across all 15 evaluated MLLMs (average accuracy <30%), while "Medium" difficulty problems showed intermediate performance. This post-hoc validation confirms that our expert assessments accurately predicted which problems would be challenging for state-of-the-art models, supporting the validity of our difficulty categorization methodology.
>
> > **[Q2] Evaluation Metrics. “The open-ended task is evaluated using exact match or by relying on an LLM-as-a-judge. However, this may not adequately capture semantic similarity or partial correctness, and alternative metrics like BERTScore could provide more nuanced assessments.", “The use of exact match as the primary evaluation metric may be too rigid and not well suited for evaluating generative responses in complex medical tasks, as evident by the 0.0% performance of several models.”, “Have the authors considered asking the clinicians to evaluate model responses?”**
>
> We thank the reviewer for raising these points about partial correctness assessment, exact match rigidity, and the need for clinician evaluations. Our evaluation deliberately employs multiple complementary metrics. While exact match (EM)  appears rigid (showing 0% for several models), this reveals a critical limitation of current MLLMs in generating precisely formatted medical responses. Higher EM scores in the in-context learning setting show that models can learn to mimic the concise formatting of in-context examples, yet still fall short of providing correct answers. We complement this with LLM-as-a-Judge for evaluating semantic correctness. We opted to utilize a binary evaluation scheme in this paper rather than measuring partial correctness for two reasons: (1) defining a notion of "partial correctness" is ambiguous in medical settings and can vary depending on the problem of interest, and (2) it is important that predictions are fully correct, as partially-correct clinical decisions can still be potentially dangerous for patients.
>
> Following your suggestion, we added human expert evaluation to complement our metrics. Due to resource constraints, we focused the human evaluation on the SMMILE dataset, recruiting 5 clinical experts who independently provided binary ratings of model responses. Each response was evaluated by 2 different clinicians in both in-context learning (ICL) and 0-shot settings. We report the expert ratings in the table below. We note that perfect inter-rater agreement (100%, κ = 1.0) was found in the ICL setting on all models, and inter-rater agreement ranging from 98.2% to 100% (κ = 0.91 to κ = 1.0) in the 0-shot setting.
>
> | Model                    | Average Expert Rating, 0-shot (%) | Expert Rating, ICL (%)* |
> |--------------------------|-----------------------------------|------------------------|
> | Claude 3.7 Sonnet        | 39.19                             | 44.14                  |
> | GPT-4o                   | 33.33                             | 43.24                  |
> | Llama-3.2-Vision-90B     | 36.94                             | 33.33                  |
> | LLaVA-v1.5-7B           | 17.12                             | 21.62                  |
> | LLaVA-v1.5-13B          | 22.52                             | 26.13                  |
> | LLaVA-NeXT-7B           | 13.51                             | 33.33                  |
> | LLaVA-OneVision-0.5B    | 19.82                             | 18.02                  |
> | LLaVA-OneVision-7B      | 25.23                             | 27.03                  |
> | LLaVA-Med-7B            | 22.52                             | 10.81                  |
> | MedGemma-4B-Multim.     | 27.03                             | 32.43                  |
> | MedVLM-R1               | 25.23                             | 24.32                  |
> | Qwen2.5-VL-3B           | 23.42                             | 23.42                  |
> | Qwen2.5-VL-7B           | 11.71                             | 27.03                  |
> | Qwen2.5-VL-32B          | 21.62                             | 32.43                  |
> | Qwen2.5-VL-72B          | 26.13                             | 31.52                  |
>
> > **[Q3] Clarification of Phrasing. “The phrasing “to quantify uncertainty in our metrics” is not precise.”**
>
> Thank you for this suggestion. We will replace the phrase “To quantify uncertainty in our metrics” with "To estimate sampling variability in our metrics" in the final version of our manuscript. Specifically, we estimate sampling variability by evaluating performance across different subsets sampled from the dataset. We employ a bootstrap resampling approach with N = 1000 bootstrap iterations. For each iteration, we randomly sample with replacement from the original results to create a simulated dataset of the same size as the original dataset, and then calculate the accuracy for this bootstrap sample. We report the mean accuracy and standard deviation across all 1000 bootstrap samples.
>
> > **[Q4] Extending LLM-as-a-Judge Approach. “The evaluation framework only employs LLaMA as the LLM judge, without justification for excluding other strong models like GPT-4 that might offer more robust judgments.”**
>
> We opted to run LLM-as-a-Judge evaluations with Llama3.3 70B because (1) Llama3.3 has been shown in prior work [1] to demonstrate strong performance on textual analysis tasks, and (2) in contrast to models like GPT-4o, Llama3.3 is open-source, generates reproducible results, and does not require payment, ensuring that our benchmark can be useful even in resource-constrained settings. In response to your suggestion, we have performed additional evaluations with GPT-4o as the LLM, and we provide results below for six models on the open-ended ICL setting:
>
> |                      | Llama3.3-as-a-Judge | GPT-4o-as-a-Judge|
> |----------------------|---------------------|------------------|
> | LLaVA-Onevision-0.5B | 21.63 ± 4.00        | 20.78 ± 3.87     |
> | LLaVA-Med            | 10.19 ± 3.06        | 10.02 ± 2.80     |
> | LLaVA-v1.5-13B       | 20.91 ± 3.49        | 22.95 ± 4.21     |
> | Qwen2.5-VL-32B       | 41.79 ± 4.73        | 41.25 ± 4.51     |
> | Qwen2.5-VL-72B       | 37.16 ± 4.46        | 35.57 ± 4.50     |
> | GPT-4o               | 49.88 ± 4.69        | 50.17 ± 4.63     |
>
> As shown in the table above, using GPT-4o-as-a-Judge results in largely similar performance to using Llama3.3-as-a-Judge. We also observe high interrater agreement between the two models (Cohen's kappa = 0.943).
>
> [1] Grattafiori et al. “The Llama 3 Herd of Models.” ArXiv.
>
> > **[Q5] Role of Medical Students. “There is ambiguity regarding the respective roles of medical students in the annotation and validation process, which could affect the reliability of the data - can they be considered as experts? Also inter-rater reliability assessment may be necessary for the subjective assessments.”**
>
> We thank the reviewer for this important clarification question. The two medical students participated exclusively in the dataset curation phase under rigorous supervision. Both medical students had extensive research experience and continuous access to medical literature throughout curation. We implemented strict quality control measures: every problem underwent manual inspection by two different senior authors and was discussed multiple times. Regarding inter-rater reliability, our human expert evaluation (conducted by 5 board-certified clinicians) achieved perfect agreement (100%, κ = 1.0) in ICL and very high agreement (98.2%-100%, κ = 0.91-1.0) in 0-shot settings, demonstrating the robustness of our evaluation framework when conducted by qualified clinical experts.
>
> > **[Q6] Details on Data Collection Platform. “The paper lacks details about the tools and technologies used to build the data collection platform, as well as information on the time it takes clinicians to construct the samples”**
>
> The front-end of our data collection platform consists of a single‑page React  client that collects each panel’s metadata (question, answer, public image URL, specialty, author, order). Once the contributor finishes a problem, the client posts the structured annotations to the back‑end. The back-end converts these annotations to an parquet file and uploads the shard to a version‑controlled Hugging Face Hub dataset. Unfortunately, we did not log creation times, because clinicians completed examples asynchronously and often in several short sessions, so any aggregate would be unreliable. Additionally, the time for clinicians to construct samples also varies widely with medical specialty and with how many in‑context examples the problem requires. We will update Section 2 of our manuscript to include these details.
>
> We again thank Reviewer GYPQ for their review of our manuscript and their positive overall assessment of our work. We hope that the above responses adequately address all concerns.

---

> > ### Comment · Reviewer_gYPq · 2025-08-07
> >
> > I would like to thank the reviewers for their detailed responses and clarifications.
> >
> > I have one remaining clarification regarding: [Q1] Difficulty of SMMILE questions.
> > Are the difficulty annotations provided with the dataset? Can you also provide this description in the paper if it's not there already?
> >
> > Otherwise I will raise my score by one point.

---

> > > ### Author Response · Authors · 2025-08-07
> > > **Clarification on dataset annotations**
> > >
> > > Yes, difficulty annotations are provided with the dataset via the flag_difficulty_llms field in our HuggingFace releases (SMMILE and SMMILE++). We will add this description to the camera-ready version if applicable. Thank you for the comment and for raising your score.

---

### Decision · Program_Chairs · 2025-09-18

**Decision:**

Accept (poster)

**Comment:**

This paper introduces a novel benchmark for multimodal in-context learning (in which the model learns to address a new task at inference time by including task examples in the prompt) in the medical domain. Clinicians created a set of 111 medical ICL problems using publicly available resourced, and state-of-the-art VLMs were evaluated on their ability to solve these problems. The work has strong real-world applications, and the provided framework includes baseline models and tools for analysis, making the benchmark potentially useful for researchers interested in both ICL and the medical domain. Additional experiments and annotations were performed in response to reviews and should be included in the next version of the paper. The paper is clearly written and reviewers’ questions and concerns were thoroughly addressed by the rebuttals. Overall, the evaluations performed provide strong empirical backing for the claims being made; significant effort went into ensuring the quality of the questions included in the benchmark and associated analysis toolkit; and the domain is important and has significant real-world applicability. Several reviewers had concerns about the exact match evaluation metric, which while convenient may not fully capture models’ performance; to some extent this is addressed by the addition of human evaluation of model responses. There is also some concern about the size of the benchmark, although this is somewhat explained by the significant expert effort that went into each problem. Broadly speaking, this began as a strong submission, and reviewers’ concerns were then well addressed with additional work that strengthens the paper.